# Epigenetic Alterations in Glioblastoma Multiforme as Novel Therapeutic Targets: A Scoping Review

**DOI:** 10.3390/ijms26125634

**Published:** 2025-06-12

**Authors:** Marco Meleiro, Rui Henrique

**Affiliations:** 1Integrated Master’s in Medicine, School of Medicine & Biomedical Sciences, University of Porto (ICBAS-UP), Rua Jorge Viterbo Ferreira 228, 4050-313 Porto, Portugal; up201905436@edu.icbas.up.pt; 2Department of Pathology and Molecular Immunology, School of Medicine & Biomedical Sciences, University of Porto (ICBAS-UP), Rua Jorge Viterbo Ferreira 228, 4050-313 Porto, Portugal; 3Department of Pathology, Portuguese Oncology Institute of Porto (IPO Porto)/Porto Comprehensive Cancer Centre Raquel Seruca (Porto.CCC), Rua Dr. António Bernardino de Almeida, 4200-072 Porto, Portugal; 4Cancer Biology & Epigenetics Group-Research Center of IPO Porto (CI-IPOP), Portuguese Oncology Institute of Porto (IPO Porto)/Porto Comprehensive Cancer Centre Raquel Seruca (Porto.CCC), Rua Dr. António Bernardino de Almeida, 4200-072 Porto, Portugal

**Keywords:** glioblastoma, epigenetics, DNA methylation, chromatin remodeling, non-coding RNA

## Abstract

Glioblastoma multiforme (GBM) is a highly aggressive primary brain tumor with a dismal prognosis despite advances in multimodal treatment. Conventional therapies fail to achieve durable responses due to GBM’s molecular heterogeneity and capacity to evade therapeutic pressures. Epigenetic alterations have emerged as critical contributors to GBM pathobiology, including aberrant DNA methylation, histone modifications, and non-coding RNA (ncRNA) dysregulation. These mechanisms drive oncogenesis, therapy resistance, and immune evasion. This scoping review evaluates the current state of knowledge on epigenetic modifications in GBM, synthesizing findings from original articles and preclinical and clinical trials published over the last decade. Particular attention is given to MGMT promoter hypermethylation status as a biomarker for temozolomide (TMZ) sensitivity, histone deacetylation and methylation as modulators of chromatin structure, and microRNAs as regulators of pathways such as apoptosis and angiogenesis. Therapeutically, epigenetic drugs, like DNA methyltransferase inhibitors (DNMTis) and histone deacetylase inhibitors (HDACis), appear as promising approaches in preclinical models and early trials. Emerging RNA-based therapies targeting dysregulated ncRNAs represent a novel approach to reprogram the tumor epigenome. Combination therapies, pairing epigenetic agents with immune checkpoint inhibitors or chemotherapy, are explored for their potential to enhance treatment response. Despite these advancements, challenges such as tumor heterogeneity, the blood–brain barrier (BBB), and off-target effects remain significant. Future directions emphasize integrative omics approaches to identify patient-specific targets and refine therapies. This article thus highlights the potential of epigenetics in reshaping GBM treatment paradigms.

## 1. Introduction

Glioblastoma multiforme (GBM) is the most aggressive and common primary brain malignancy, with an incidence rate ranging from 0.59 to 5 cases per 100,000 individuals. This rate has been rapidly increasing in many countries due to factors such as aging populations and advancements in diagnostic precision [1,2]. The standard treatment, established by the Stupp protocol, involves surgical tumor resection with broad margins, followed by radiotherapy and concomitant adjuvant temozolomide (TMZ) [3]. Other approved treatments for GBM include lomustine, carmustine, bevacizumab, carmustine wafer implants, and tumor-treating fields, which are mostly utilized to manage recurrent GBM and alleviate its symptoms [4].

Despite these treatments, median survival for GBM patients is approximately 15 months, and the 5-year relative survival rate is only 6.8% (although dependent on sex and age at diagnosis). Progression of GBM entails severe neurological decline, with physical and cognitive impairments, significantly reducing patients’ quality of life and bringing emotional and financial challenges to caregivers. Tumor recurrence and drug resistance remain significant challenges, contributing to the poor prognosis and limiting the effectiveness of available therapies [1,2,5].

Comprehensive genome-wide studies have cataloged somatic mutations, copy number variations, and genetic rearrangements while also revealing transcriptomic, epigenomic, proteomic, and metabolomic profiles of GBM [6,7,8,9,10]. These multiomics approaches have also illuminated the role of key cellular processes such as autophagy, linking it to oncogenic signaling and cellular stress responses [11,12,13].

Technological advances, such as next-generation sequencing, RNA expression analyses, and DNA methylation profiling, have reshaped our understanding of central nervous system (CNS) tumors, creating the need for novel and updated classification systems. After the 2016 version introduced specific molecular alterations as diagnostic criteria for the first time, the 5th World Health Organization (WHO) classification system carried on refining the stratification of existing tumors while still including key histological and immunohistochemical diagnostic features. The 2021 scheme divides CNS tumors into six major categories, as summarized in Figure 1.

Hence, “GBM, isocitrate dehydrogenase (IDH)-wildtype” is defined as a grade 4 adult-type diffuse astrocytoma lacking IDH and histone mutations, with simultaneous gain of chromosome 7 and loss of chromosome 10, epidermal growth factor receptor (EGFR) amplification, TERT promoter mutations, mitoses, necrosis, or microvascular proliferation. Other frequently observed molecular abnormalities include O^6^-methylguanine-DNA methyltransferase (MGMT) promoter methylation, CDKN2A/B deletions, phosphatase and tensin homolog (PTEN) alterations, TP53 mutations, MDM2/4 amplifications, and BRAF V600E mutations [4,14,15]. Nonetheless, in IDH-wildtype diffuse astrocytomas occurring in younger patients, it is important to consider the distinct subtypes of diffuse pediatric-type gliomas—adult-type tumors can rarely be found in children, particularly young adults, while pediatric-type tumors may occasionally occur in adults [16].

Moreover, “Astrocytoma, IDH-mutant” is a diffuse astrocytoma with activating mutation in IDH1 or IDH2 (as well as altered ATRX, TP53, or CDKN2A expression) that is graded 2-4 within type, abolishing the term “GBM, IDH-mutant”. The gain-of-function mutation in IDH1/2 leads to overproduction of the oncometabolite D-2-hydroxyglutarate and inhibition of enzymes that require α-ketoglutarate as a cofactor, such as DNA demethylases, causing genomic cytosine-phosphate-guanine (CpG) hypermethylation and impairing cellular differentiation [17,18]. The IDH1 mutation is often considered an initiating event in tumorigenesis, being present in all tumor cells, making it an attractive therapeutic target, including for vaccine development [19,20]. IDH mutations also affect the splicing and expression of epigenetic modifiers such as DNA methyltransferases and methylation readers [21]. Although its standard treatment is similar to that of IDH-wildtype GBM, it exhibits slower progression and a more favorable prognosis [14].

Given the profound complexity of GBM, overcoming this malignancy requires multifaceted therapeutic approaches. These may include combining surgical tumor debulking with strategies targeting molecular drivers, such as engineered immunomodulators with high specificity and cytotoxicity or oncolytic viruses [6]. Leveraging insights from epigenetics, coupled with precision medicine tools like omics-driven biomarker identification, may enable the development of more effective and personalized treatment strategies for GBM [22,23,24]. This scoping review explores the evolving role of epigenetic alterations as therapeutic targets, biomarkers, and combinatory strategies in the fight against GBM, with an emphasis on bridging preclinical discoveries and clinical applications.

## 2. Methods

The protocol was developed according to the PRISMA Extension for Scoping Reviews (PRISMA-ScR) recommendations and was retrospectively registered with the Open Science Framework (OSF; Center for Open Science, Washington, DC, USA) on 4 June 2025 [25]. It is publicly accessible at: https://doi.org/10.17605/OSF.IO/9HARY. This scoping review was guided by the PCC framework, focusing on GBM patients (Population), epigenetic alterations as therapeutic targets (Concept), and the limited effectiveness of current standard treatments (Context) (Appendix A).

All relevant preclinical studies and clinical trials—single-arm or double-arm, including both randomized and nonrandomized controlled trials—addressing epigenetic alterations in GBM as therapeutic targets were eligible for inclusion. Studies were excluded if they did not clearly define their methods and/or results. Only studies published in English and within ten years (2014–2024) were considered. Editorials, conference abstracts, books or book chapters, case reports, case series, literature reviews, meta-analyses, and preprints were excluded.

We conducted searches in the databases PubMed, Scopus, and Web of Science from inception until 16 October 2024. ClinicalTrials.gov was also searched from inception to 22 December 2024 for ongoing or completed registered trials.

The search strategy included the terms “glioblastoma”, “epigenetic”, and “therapeutic targeting”, along with MeSH terms, synonyms, and appropriate filters. The final PubMed search string was as follows: “(Glioblastoma epigenetic therapeutic targeting) NOT (review[pt])”, with filters applied for full text, English language, publication within the last 10 years, and exclusion of preprints. A manual search, primarily through backward citation tracking, was also conducted to identify additional relevant studies.

All search results were exported into EndNote 21.5 (Clarivate, Philadelphia, PA, USA). Duplicate records were removed using EndNote, initially by matching records with the same title and author published in the same year, followed by a second pass using title, author, and journal. This process identified 93 duplicates. An additional 78 duplicates were identified through manual review. A total of 171 duplicates were removed before screening. Reference screening was performed in two stages. Initially, titles and abstracts were screened by one author against the eligibility criteria. Full texts were retrieved for all potentially relevant studies. One author then reviewed the full texts and screened the trial registry entries for inclusion, with verification by the second author. Discrepancies were resolved through discussion and consensus. The study selection process is illustrated in Figure 2.

Data charting was conducted using a standardized form by one author and verified by the other. Information extracted from each study included publication year, model system used (e.g., in vitro or in vivo), epigenetic targets, therapeutic interventions, stage of development (preclinical or clinical), and any reported study limitations. No critical appraisal of included sources was performed, in accordance with standard scoping review methodology. The data charting process and results are summarized in Table A1, Table A2 and Table A3.

## 3. Results

A total of 547 records were identified through databases and manual searches. After removing duplicates, 376 unique records were screened by title and abstract, with 311 progressing to full-text review. Ultimately, 286 studies met the inclusion criteria and were included in the final analysis. The study selection process is detailed in the PRISMA-ScR flow diagram (Figure 2).

The included studies comprised both preclinical and clinical investigations exploring epigenetic alterations in GBM as therapeutic targets. Charted data (Table A1, Table A2 and Table A3) reflects a diverse range of epigenetic mechanisms, including DNA methylation, histone modifications, and ncRNAs. Therapeutic approaches varied, encompassing small-molecule inhibitors, RNA-based therapies, and gene-editing strategies. Most included studies were conducted in preclinical settings using established GBM cell lines, patient-derived xenografts, or animal models, with a smaller proportion representing early-phase or small-scale clinical trials.

The synthesis of findings was primarily narrative, focusing on the translational relevance of the identified strategies. Evidence was prioritized based on its current or potential applicability to clinical practice. Building on the broader body of research in the field and particularly the comprehensive 2020 review on epigenetics in GBM as a gateway to therapeutic development [26], this review emphasizes studies published between 2021 and 2024 to highlight recent advances and evolving trends.

## 4. Discussion

### 4.1. Glioblastoma Multiforme: An Overview

GBM is a highly heterogeneous cancer with invasive behavior, making complete surgical removal nearly impossible. Despite combining radiotherapy with TMZ chemotherapy, treatment often fails because GBM cells develop resistance to both therapies over time. Histopathological features such as necrosis, cellularity, angiogenesis, and mitosis may not accurately explain the failure of treatment and poor clinical outcomes [27]. An illustrative depiction of the epigenetic and signaling pathways involved in GBM pathogenesis and their therapeutic targeting is presented in Figure 3.

A major challenge in GBM treatment is its extensive cellular and molecular heterogeneity. GBM harbors a small population of glioma stem-like cells (GSCs), which are multipotent, capable of self-renewal, and especially resistant to chemoradiation [28,29]. GSCs participate in abnormal angiogenesis, synapse formation, invasiveness, myelination, and metabolic alterations. Intriguingly, pathways critical to normal embryogenesis are repurposed during gliomagenesis, with neurotransmitters like dopamine, noradrenaline, and glutamine in the microenvironment exploited by tumors to support growth and survival. In fact, metabolic adaptation and reprogramming are crucial in gliomagenesis, with glutamine starvation enhancing the expression of enzymes such as PSAT1, SHMT2, and MTHFD2, the latter of which emerges as a promising therapeutic target due to its elevated levels in GBM. Serine synthesis, driven by autophagy rather than glycolysis, supports this process, and inhibiting either autophagy or MTHFD2 significantly impairs GBM cell survival and growth [30]. Immune pathways are also differentially regulated at various stages of gliomagenesis, including the downregulation of antigen presentation genes, reflecting the tumor’s adaptability [6,31,32]. This hierarchical cell organization enables a pattern of relapse and progression in GBM, since residual GSCs can survive treatment and contribute to tumor recurrence. Efforts have focused on identifying specific markers for GSCs, such as OCT4, NANOG, SOX2, and CARM1, to address the challenge of therapeutic resistance [31,33].

A study confirmed the spatial heterogeneity and suggested evolutionary trajectories of GBM using a three-dimensional spatial sampling approach, combining surgical neuronavigation with pre-operative magnetic resonance imaging (MRI) scans. Early tumor evolution is marked by CDKN2A/B loss and EGFR, platelet-derived growth factor receptor A (PDGFRA), and CDK4 gains. Nearly 40% of patients showed subgroup heterogeneity, with spatially distinct tumor regions deviating from dominant subtypes, and homogeneous subtypes correlated with longer median survival (477 vs. 293 days). There was also IDH1 differential expression within tumors, with higher expression in the cellular tumor regions and lower levels at the invasive tumor edge. Gain of chromosome 7, containing EGFR, PTPRZ1, and PTN, possibly locks cells into an undifferentiated intermediate progenitor cell state. Additionally, spatial analysis revealed distinct microenvironments at the GBM core and periphery. These microenvironments were shaped by the distribution of nonmalignant cell types and tumor cell-intrinsic programs activated in response to regional factors like injury and hypoxia. Notably, neuronal hijacking occurred at the tumor periphery, where tumor cells interacted with neurons. In the tumor core, different immune microenvironments were observed, including immune-hot areas with interferon signaling and T cell infiltration and immune-cold areas undergoing mesenchymal differentiation, coordinated by the master regulator AP-1 [32,34,35,36]. Other techniques, like GSC-derived GBM cerebral organoid models, have confirmed this epigenetic heterogeneity and dynamic chromatin changes mirroring early neural development. This also allows cell-state distribution manipulation by targeting chromatin regulators, which could enable “state-selective lethality”, a therapeutic strategy that drives cells into drug-sensitive states. An alternative strategy is targeting WDR5, crucial for GSC self-renewal and tumor initiation, by disrupting the epigenetic maintenance of GSCs and circumventing the challenges of directly targeting transcription factors like SOX2 [37,38].

The blood–brain barrier (BBB) is another main challenge to effective treatment, blocking the uptake of over 98% of systemic therapies. To penetrate the BBB, agents must meet specific physiochemical criteria, such as lipophilicity, charge, and size, which complicates drug development. Convection-enhanced delivery offers a potential solution by directly distributing drugs into brain tissue using mechanical force. Nanoparticle (NP) drug delivery systems offer promising advancements for glioma therapy, with over 500 clinical trials already exploring NP-based treatments for various diseases [39].

Epigenetic alterations have long been recognized as key contributors to gliomagenesis and tumor progression. Epigenomics examines reversible chemical modifications to DNA, histone proteins, and messenger RNA (mRNA) that influence gene expression without altering the genetic sequence. These modifications include changes to histone proteins, such as acetylation, methylation, and ubiquitination, which may either activate or suppress transcription. DNA methylation, particularly at CpG sites, generally inhibits gene expression by blocking the binding of transcription factors to promoter or enhancer regions. While methylation patterns are tightly controlled during development and in normal tissues, they are often disrupted in pathological conditions [6].

Non-coding RNAs (ncRNAs) also play a significant role in this regulatory landscape. MicroRNAs (miRNA, miR) bind to the 3′ untranslated regions of mRNA, blocking translation or promoting degradation. Although primarily associated with gene silencing, some miRNAs can enhance transcript stability and translation, increasing gene expression. Notably, over half of miRNA-encoding genes are located in genomic regions frequently altered in tumors, suggesting their significant role in regulating oncogenes and tumor suppressor genes [40,41,42]. Long non-coding RNAs (lncRNAs, non-protein-coding RNA molecules over 200 nucleotides in length) regulate gene expression by influencing chromatin structure, interacting with nuclear molecules, and serving as precursors or sponges for microRNAs. They play crucial roles in development and cancer progression, acting as both activators and repressors of transcription [43].

### 4.2. DNA Methylation Modulation

DNA methylation is a key epigenetic mechanism that regulates chromatin structure and gene expression. Cancer cells often exhibit global DNA hypomethylation paired with hypermethylation at specific loci, contributing to genomic instability and silencing of tumor suppressor genes, respectively. In gliomas, DNA hypomethylation correlates with aggressive tumor behavior and poor prognosis [44].

DNA methylation is controlled by the activities of DNA methyltransferase (DNMT1), which maintains methylation patterns, and de novo DNA methyltransferase (DNMT3A/3B), which establishes de novo methylation. Although mutations in these methyltransferases are not common in brain tumors, mutations or amplification in receptor tyrosine kinase (RTK) genes may drive abnormal DNA methylation patterns in gliomas, encompassing metabolic reprogramming and contributing to the distinct hypomethylation phenotype observed. Mechanistic target of rapamycin complex 2 (mTORC2) plays a role in the DNA hypomethylator phenotype in GBM, particularly in tumors with platelet-derived growth factor receptor (PDGFR) amplification or the gain-of-function EGFR mutation, EGFRvIII. mTORC2 integrates aberrant RTK signaling with environmental nutrient levels to regulate histone modifications and promote tumor growth, driving the hypomethylator phenotype by epigenetically regulating DNMT3A, which in turn remodels the tumor-supportive glutamate metabolism network [45].

TMZ is an alkylating chemotherapeutic agent that induces cytotoxicity and apoptosis by alkylating DNA at numerous sites. MGMT can remove alkyl groups from the O^6^ position of guanine, repairing the most toxic event caused by TMZ and leading to chemoresistance [28,46]. In around 40% of GBM patients, epigenetic modifications of the CpG island at specific CpG sites within the MGMT promoter result in gene silencing, impairing the repair of DNA alkylation and thereby improving response to TMZ. Accordingly, MGMT epigenetic silencing has been associated with longer survival, and testing its methylation status has become one of the most relevant biomarkers for prognosis and therapeutic guidance [47,48,49,50]. Nevertheless, other factors may play a role, as IDH1-wildtype GBM demonstrates a more significant association with MGMT status than IDH1-mutated gliomas [51,52].

ADP-ribosylation factor-like protein 13B (ARL13B), regulated by Enhancer of Zeste Homologue-2 (EZH2), interacts with inosine-5′-monophosphate dehydrogenase 2 (IMPDH2), a key enzyme in purine biosynthesis. This interaction hinders the purine salvage pathway, contributing to TMZ resistance. Disrupting IMPDH2 activity with the FDA-approved drug mycophenolate mofetil significantly improved TMZ efficacy and extended survival in patient-derived xenograft mouse models of GBM. Targeting IMPDH2 with mycophenolate mofetil offers a clinically viable strategy to enhance the effectiveness of alkylating agents in GBM [53]. Additionally, GBM cells utilizing anaerobic glycolysis, even under normoxic conditions, contribute to TMZ resistance. Suppressing HIF-1α may increase TMZ sensitivity in GBM cells and potentially reduce the required therapeutic dose when combined with KC2F7, a specific inhibitor of HIF-1α. Targeting metabolism in this way enhances the cytotoxic and apoptotic effects of TMZ, offering potential strategies for improving standard GBM treatment by regulating key metabolic pathways [54].

Endosomes play a vital role in regulating RTK signaling by modulating its intensity, timing, and spatial localization. RTK signaling is a key pathway frequently altered in GBM, with genetic changes present in 86% of tumors, including mutations in EGFR, ERBB2, and PDGFR. Sorting nexin 10 (SNX10) is a key regulator of endosomal sorting specific to PDGFRβ in GSCs, a pathway critical to GBM initiation and progression [55]. Despite the central role of PDGFR and related RTKs in GBM pathogenesis, clinical trials with multitargeted kinase and PDGFR inhibitors, such as imatinib, tivozanib, and nilotinib, failed to extend survival [56,57,58,59]. However, SNX10’s role in regulating PDGFRβ suggests that it might serve as a biomarker to predict responses to PDGFR inhibitors, enabling better patient stratification. Additionally, therapeutic inhibition of SNX10 or related signaling pathways, such as STAT3, could provide novel strategies to target GBM vulnerabilities [55].

Epidermal growth factor (EGF) is essential for cell growth, proliferation, and differentiation. Its receptor, EGFR, is often overexpressed and persistently activated in GBM, driving progression via pathways, including RAS/RAF/MEK/ERK, PI3K/AKT/mTOR, Src kinases, and STAT transcription factors. Pyruvate kinase M2 (PKM2), a key enzyme regulating the final step of glycolysis, plays a central role in the Warburg effect. EGF-induced ERK1/2 activation stimulates PKM2 phosphorylation at serine 37 in GBMs, while EGF also promotes phosphorylation of O-GlcNAc transferase (OGT) at Y976, enhancing its binding to PKM2. This increases PKM2 O-GlcNAcylation and de-tetramerization, reducing its activity. Additionally, OGT phosphorylated at Y976 preferentially recognizes other phosphotyrosine-binding proteins, such as STAT1, STAT3, STAT5, PKCδ, and p85, indicating a broader role in substrate selection. These findings highlight O-GlcNAcylation as a promising therapeutic target in EGFR-expressing cancers [60].

Serotonin has been implicated in glioma progression, promoting tumor cell proliferation, migration, and invasion, although its effects on tumor growth remain contradictory. Expression of serotonin receptor (5-HT_7_R) in a *Drosophila* glioma model, which mimics human glioma via co-expression of active PI3K and EGFR in glial cells, reduced larval lethality, restored normal brain morphology, and reversed molecular markers altered in gliomas. The tumor-suppressive effects of 5-HT_7_R likely involve modulation of cAMP pathways, metabolic changes, and interference with EGFR signaling. These findings position 5-HT_7_R as a promising therapeutic target for glioma, with the *Drosophila* model serving as a platform for drug screening and the development of 5-HT_7_R-targeted treatments [61].

Ubiquitin-specific peptidase 6 N-terminal-like (USP6NL), a GTPase-activating protein, plays a critical role in regulating EGFR trafficking and signaling in GBM. Overexpression of USP6NL stabilizes EGFR, enhancing AKT signaling, tumor cell survival, and therapy resistance. In TMZ-resistant GBM cells, USP6NL promotes self-renewal, invasion, and epithelial-mesenchymal transition (EMT), while knockdown of USP6NL sensitizes cells to TMZ. Mechanistically, USP6NL modulates the ubiquitin-proteasome system, which governs protein degradation and cellular homeostasis. Targeting USP6NL and the ubiquitin-proteasome system offers a novel therapeutic strategy to overcome resistance and recurrent GBM, potentially disrupting key survival pathways and enhancing treatment efficacy [62].

The Mre11-Rad50-NBS1 (MRN) complex plays a critical role in DNA damage repair by initiating homologous recombination (HR) and non-homologous end joining pathways for double-strand breaks (DSBs). Retinoblastoma binding protein 4 (RBBP4), a component of chromatin-modifying complexes such as Polycomb Repressive Complex 2 (PRC2), NuRD, and SIN3, regulates chromatin assembly during normal replication and DNA damage repair. Additionally, RBBP4 enhances MRN complex expression, promoting DNA repair and contributing to resistance to TMZ and radiation in MGMT-negative GBM. Targeting RBBP4 or the MRN complex may provide new therapeutic strategies for overcoming resistance and improving treatment outcomes, mainly in MGMT-negative GBM [63].

Along with MGMT gene methylation, retinoic acid receptor β (RARβ) promoter methylation is also positively correlated with treatment responses to radiotherapy and chemotherapy in GBM, suggesting RARβ-targeting therapies as potential candidates (including the FDA-approved drugs sulfanilamide, fulvestrant, sulfamerazine, bacampicillin, myricetin, and fursultiamine) [64].

Unlike the well-studied 5-methylcytosine, 5-hydroxymethylcytosine (5hmC) is an epigenetic modification predominantly found in enhancers, gene bodies, and promoters of actively transcribed genes. The depletion of 5hmC has been linked to hypermethylation of gene bodies in various cancers, including grade 4 gliomas. Patient survival may be predicted by 5hmC-based prognostic models, potentially surpassing traditional markers like IDH1 mutations, and may help identify new therapeutic targets [65].

The transcription factor SOX2 drives GSC induction and malignancy in IDH1/2 wild-type GBM. SOX2 represses the TET2 demethylase, resulting in reduced levels of 5hmC, a key DNA modification associated with tumor suppression. The loss of TET2 and 5hmC enhances GSC self-renewal and tumorigenic capacity. Furthermore, SOX2 activates the onco-miR miR-10b-5p, which directly targets and represses TET2, contributing to a hypermethylated and oncogenic phenotype in GSCs. Targeting SOX2-miR-10b-5p-TET2 signaling to counteract the hypermethylated state and malignancy of GSCs in GBM could be an efficient therapeutic strategy [66].

### 4.3. Histone Modification

Histone acetyltransferases (HATs) and histone deacetylases (HDACs) are key enzyme families with opposing roles in regulating histone acetylation—while HATs add acetyl groups to histones, promoting active transcription, HDACs remove these groups, leading to gene silencing. Unbalanced histone acetylation is linked to tumorigenesis, since reduced histone acetylation suppresses tumor-regulatory genes [67]. Besides having prognostic significance, histone acetylation is linked to differential expression of immune checkpoint genes, supporting the development of precision therapies targeting histone acetylation and the immune microenvironment [68,69].

HDAC2 knockdown induces GBM cell death by controlling miR-3189 expression, repressing glucose transporter 3 (GLUT3) transcription, and regulating glucose metabolism. This suggests that targeting HDAC2 could restore drug sensitivity in GBM [70]. HDAC2 is also an epigenetic regulator of GSCs across various genetic backgrounds, working with transforming growth factor β (TGF-β) pathway proteins, specifically SMAD3 and SKI, to regulate chromatin organization and gene expression. These chromatin modifications affect the expression of key genes such as SMAD3 and SOX2, which are vital for GSC self-renewal and growth. Additionally, HDAC2’s protein–protein interaction with SMAD3-SKI promotes cell cycle progression and suppresses genes related to cell fate specification. These findings suggest that disrupting the HDAC2-SMAD3-SKI pathway with specific inhibitors could be an effective therapeutic strategy for targeting the drug-resistant GSC population in GBM [71].

HDAC1, linked to higher tumor grades and worse prognosis, has a nonredundant critical involvement in the proliferative potential of GSCs. The loss of HDAC1 in GSCs leads to a decrease in key glioma stemness markers and stabilizes p53, inducing cell death, with no compensation from its paralogue HDAC2. This contrasts with normal neural stem cells, where HDAC1 is dispensable, strengthening the need for isoform-specific HDAC inhibitors (HDACis) for targeted therapies in GBM [72].

SLC30A3, a factor regulated by a super-enhancer (SE), functions as a tumor suppressor and exhibits significantly lower expression in GBM. Its activity is negatively regulated by HDAC1 through reduced H3K27ac levels, which impairs MAPK signaling. Targeting the HDAC1/SLC30A3/p38 MAPK axis emerges as a potential therapeutic strategy, highlighting SLC30A3’s role in curbing GBM malignancy via MAPK pathway activation [73].

HDACi eliminates cancer cells by inducing the expression of cell cycle repressors and pro-apoptotic genes [74]. Early studies highlighted the potential of HDACis like valproic acid in improving survival outcomes for glioma patients. Subsequent research, although encouraging, has yielded mixed results. Interestingly, HDAC inhibitors have shown heightened efficacy in gliomas harboring the IDH1 R132H mutation [19].

Among them, belinostat (PXD-101), a pan-HDACi with enhanced BBB penetration, has shown promising results in a pilot clinical trial, where newly diagnosed GBM patients received belinostat combined with concurrent radiotherapy and TMZ [75]. Belinostat improves upon earlier HDACis, such as vorinostat (suberoylanilide hydroxamic acid, SAHA), with greater potency and efficacy, particularly in restoring brain metabolite levels like N-acetylaspartate and myo-inositol, as shown by spectroscopic MRI. Results indicated improved six-month progression-free survival (84% vs. 54%, *p* = 0.073) and increased median overall survival by approximately 2.7 months compared to the control group. Voxel-based analysis of recurrence patterns suggested that belinostat may act as a radiosensitizer, improving in-field tumor control and potentially shifting recurrence patterns out-of-field [76]. Engineered IDH1-mutant glioma cell lines exhibited increased sensitivity to HDACis, including belinostat, and disclosed enhanced apoptotic responses. Radiographic and spectroscopic MRI analyses from the clinical trial support these findings, highlighting superior responses in IDH1-mutant GBM compared to IDH-wildtype cases [77].

Givinostat is another pan-HDACi inhibitor with BBB penetration, which has shown antitumor activity in GSC models and whose safety in humans has been established through phase I/II clinical trials for polycythemia vera. Mechanistically, givinostat inhibits Sp1 expression, a transcription factor critical for MGMT expression, impairing Sp1’s DNA-binding activity through HDAC-dependent mechanisms, thus attenuating MGMT-dependent resistance. These findings indicate that givinostat does not compromise TMZ sensitivity in GSCs lacking MGMT, supporting its utility across different GSC populations and the combination of givinostat with TMZ as a novel therapeutic strategy [46].

Another HDACi, trichostatin A (TSA), exhibits anticancer properties by inhibiting HDAC6, promoting cell death through mechanisms such as DNA repair suppression, and acting as a radiosensitizer. However, the TSA and TMZ combination showed antagonistic effects in GBM cell lines with distinct epigenetic patterns of TMZ resistance, with stronger antagonism observed in cells with higher MGMT expression. This suggests that MGMT expression levels influence the efficacy of TSA and TMZ combinations, emphasizing the importance of the MGMT gene over mismatch repair genes in mediating TMZ resistance [78].

The selective HDACi LMK235 demonstrated significant antitumor effects on GBM cells by reducing cell viability and colony formation through apoptosis and autophagy pathway induction. Transcriptomic analysis identified SCNN1A as a gene significantly downregulated by LMK235 treatment, and SCNN1A silencing further reduced GBM cell viability. These results suggest that targeting HDAC4/HDAC5 via LMK235 and modulating SCNN1A expression may offer a promising therapeutic strategy for GBM [79].

Domatinostat (4SC-202), a class I HDAC inhibitor, preferentially inhibits GSCs over their differentiated counterparts, selectively impairing GSC survival and self-renewal without significantly affecting normal fibroblasts or differentiated GSCs at effective concentrations. Hence, domatinostat is a potential anti-GSC therapy, offering a promising avenue for preventing relapses in GBM patients [80].

EZH2, the catalytic subunit of PRC2, is primarily associated with histone methylation, as well as promoting DNA methylation marks (by recruiting DNMTs). Specifically, EZH2 mediates the trimethylation of lysine 27 on histone H3 (H3K27me3), a well-known marker of transcriptional repression. EZH2 altered expression has been implicated in EMT across various tumors, and when upregulated in GBM, it is associated with worse prognosis [40,41,42,81,82].

EZH2 also interacts with HP1BP3, a heterochromatin-related protein, through a PRC2-independent mechanism to promote proliferation, stemness, and TMZ resistance in GSCs. Specifically, the EZH2-HP1BP3 complex enhances WNT7B expression by reducing the repressive histone marker H3K9me2, potentially disrupting HP1-HDAC interactions. The WNT/β-catenin pathway, which influences tumor growth, angiogenesis, invasiveness, and therapeutic resistance, emerges as a critical mediator in this process. Pharmacological inhibition of WNT7B effectively reverses TMZ resistance in HP1BP3-overexpressing GBM cells. Therefore, targeting the EZH2-HP1BP3 axis could be a promising therapeutic strategy for overcoming GSC-driven resistance in GBM [81].

Notably, EZH2 inhibitors, such as GSK126, promote apoptosis in GBM cells by inducing autophagy via the EZH2/miR-101/mTOR signaling axis—EZH2 knockdown upregulates miR-101, which regulates EZH2 in a positive feedback loop, further enhancing autophagy-mediated apoptosis. By disrupting the EZH2/miR-101/mTOR feedback loop, novel treatments could be developed to inhibit GBM progression and improve patient outcomes [42]. While EZH2 inhibitors like tazemetostat have shown promise in clinical trials, their effectiveness in GBM remains inconsistent, since prolonged inhibition of EZH2 in GBM cells has been linked to tumor progression due to activation of cell proliferation and DNA damage repair pathways. Ribosomal S6 kinase 4 (RSK4) is a serine/threonine kinase highly expressed in GBM, being associated with poor prognosis and enriched in GSCs. RSK4 directly phosphorylates EZH2 at S21, activating the EZH2/STAT3 pathway through a PRC2-independent mechanism and promoting GSC maintenance, while the knockdown of RSK4 restored sensitivity to EZH2 inhibitors and reduced GSC properties. Combination therapy targeting RSK4 (with the inhibitor BI-D1870) and EZH2 significantly inhibited tumor progression and improved survival in an orthotopic xenograft model [82].

Epigenetic mechanisms, beyond CpG methylation, may be involved in regulating the Wnt pathway in GBM, modulating the expression of key Wnt markers. Wnt5a, Fzd-2, β-catenin, and Wnt3a protein levels are generally increased in GBM compared to normal brain tissues. Expression of Fzd-2 and Wnt3a is significantly higher in recurrent GBM samples, while Wnt7b shows a marked decrease in these recurrent cases. As for DNA methylation, Wnt3a and Wnt7b have notably higher methylation levels in GBM compared to other markers, with Fzd-10 displaying the highest levels of methylation overall. This inverse relationship between methylation and protein expression is evident in several markers but not in the case of Wnt7b, where decreasing methylation levels do not correlate with increased protein expression. The intricate regulation of Wnt signaling in GBM highlights the potential of targeting this pathway for therapeutic strategies [83].

Elevated pleckstrin homology domain containing A4 (PLEKHA4) expression, driven by DNA hypomethylation, is linked to tumor progression, WNT signaling, immune cell infiltration, and stemness maintenance. Knockdown experiments showed reduced glioma cell proliferation, and drug sensitivity analysis suggests potential for targeting PLEKHA4 with kinase inhibitors [84].

The AKT/mTOR signaling pathway is a key regulator of cellular processes, including apoptosis inhibition, tumor progression, and chemoresistance. PTEN, a tumor suppressor, plays a crucial role in this pathway by dephosphorylating phosphatidylinositol 3,4,5-trisphosphate, thereby antagonizing PI3K/AKT signaling and maintaining cellular homeostasis [85,86]. Bevacizumab, a monoclonal antibody targeting vascular endothelial growth factor (VEGF)-A, inhibits the AKT/mTOR pathway, inadvertently activating autophagy as a survival mechanism. While bevacizumab effectively reduces proliferation and enhances apoptosis by modulating pro- and anti-apoptotic protein levels, its therapeutic impact is limited by this compensatory autophagic response. Blocking autophagy with chloroquine significantly amplifies bevacizumab-induced apoptosis, suggesting that autophagy enables GBM cells to tolerate antiangiogenic stress. These findings highlight the interplay between the AKT/mTOR pathway and autophagy in driving resistance and point to autophagy inhibition as a promising strategy to enhance bevacizumab efficacy in GBM treatment [87].

Polycomb group (PcG) proteins, key regulators in PRC1 and PRC2 complexes, influence cell proliferation and tumorigenesis. Chromobox (CBX) family members, essential components of PRC1, contribute to epigenetic regulation by targeting PRC1 to chromatin. Analysis revealed elevated expression of CBX2/3/5/8 and reduced expression of CBX6/7 in GBM, with correlations to tumor grade and recurrence. Overexpression of CBX3/8 and underexpression of CBX6 were linked to shorter survival. Functional assays confirmed CBX8’s role in promoting glioma cell proliferation. CBX7 showed increased methylation consistent with its reduced expression. CBX gene expression was also associated with immune cell infiltration, particularly CD4^+^ T cells [88].

Additionally, elevated CBX2 expression is strongly associated with higher pathological grades and TMZ resistance. Functional studies revealed that CBX2 recruits EZH2 to mediate H3K27me3 modifications on the PTEN promoter, silencing its expression. Conversely, CBX2 knockdown inhibits tumor growth and sensitizes cells to chemotherapy by restoring PTEN activity and suppressing the AKT/mTOR pathway. These findings highlight CBX2 as a critical driver of glioma progression and a potential therapeutic target [86].

Succination of PTEN, a newly identified posttranslational modification, prevents its localization on the cell membrane, reducing its inhibitory effects on the PI3K/AKT pathway. Whereas PTEN is frequently mutated in GBM, it remains intact in many GSCs. PTEN’s regulation in GSCs appears to depend on its interaction with MMS19, a critical protein involved in iron–sulfur (Fe-S) cluster assembly (CIA). PTEN disrupts MMS19-dependent CIA machinery, which is crucial for DNA metabolism and repair. In GSCs, the purine synthesis pathway fuels fumarate to succinate PTEN, which weakens PTEN’s interaction with MMS19, promoting GSC maintenance and tumorigenesis. This suggests that the tumor-suppressive function of PTEN is masked in GSCs, which in turn offers therapeutic opportunities to reactivate PTEN in cancer treatment [85].

Stearoyl-CoA desaturase (SCD), an enzyme regulating membrane fluidity by desaturating fatty acids, is implicated in tumorigenesis, but resistance to its inhibition presents a challenge in cancer therapy. In GBM, SCD is co-deleted with PTEN on chromosome 10 and exhibits variable methylation patterns. These epigenetic alterations contribute to resistance, even with remaining SCD activity, with a universal resistance mechanism driven by FOSB-mediated signaling resulting in target overexpression. Therefore, SCD expression and methylation status could serve as key biomarkers for patient stratification in clinical trials evaluating SCD inhibitors [89].

MK2206, an allosteric inhibitor of phosphorylated AKT, has shown promise in GBM therapy by enhancing the effects of existing treatments like gefitinib and effectively blocking AKT phosphorylation. AKT phosphorylation is closely linked to EGFR activity, particularly in EGFR/EGFRvIII-positive GBMs, where EGFR amplification frequently drives PI3K signaling. EGFR/EGFRvIII activation silences AJAP1, an antioncogene known to suppress cell invasion by reorganizing the F-actin cytoskeleton and reducing filopodia extensions. Thus, loss of AJAP1 expression, observed in over 80% of primary GBM cases, was shown to enhance tumor cell invasion. Given the limited success of EGFR tyrosine kinase inhibitors in some GBM patients, MK2206 represents a promising targeted therapy by addressing the underlying AKT-driven epigenetic changes and cytoskeletal remodeling associated with GBM invasiveness [90].

Prostaglandin F2 receptor inhibitor (PTGFRN), a transmembrane cell adhesion molecule (CAM) of the immunoglobulin superfamily, is upregulated in GBM, promoting cell growth and radiation resistance through the PI3K-AKT signaling pathway. Approximately 35% of CAMs show differential regulation in GBM, exhibiting both oncogenic and tumor-suppressive functions. Importantly, over half of the deregulated CAMs were found to be miRNA targets, suggesting a significant role for miRNA-mediated regulation in CAM expression. PTGFRN was experimentally validated to exhibit pro-migratory and pro-proliferative functions in GBM, highlighting its role in tumor invasion [91].

Connexin 43 (CX43), a key gap junction protein, mediates both autocrine and paracrine signaling pathways crucial for tumor growth. The Sonic Hedgehog (SHH) pathway, known for its role in cell network signaling, may regulate CX43 expression, influencing GBM pathobiology. SHH-GLI signaling fosters intercellular communication and microenvironment changes, promoting tumorigenesis, and SHH signaling supports the stemness of GBM by influencing CX43-mediated cellular interactions that maintain the GSC population. These findings suggest that targeting the CX43-SHH axis may offer new therapeutic strategies for GBM [92].

Lamellipodin, a cytoskeleton-associated protein, was found to cooperate with EGFR and interact with RICTOR, a key component of the mTOR pathway, to regulate GBM invasion and radiosensitivity. Lamellipodin is involved in clonogenic radiation survival and cell proliferation, which underscores its complex role in EGFR signaling and its potential as a therapeutic target in cancer treatment [93].

NF1 (a GTPase-activating protein) mutation results in RAS-GTP persistence, triggering the MEK/ERK phosphorylation cascade and activating the MAPK pathway, driving cell growth and survival. Additionally, RAS-GTP interacts with PI3K, enhancing its kinase activity and activating the downstream AKT-p65 signaling pathway, which further supports cell proliferation, resistance to apoptosis, and chemotherapy resistance. In NF1-mutated GBM with an active RAS pathway, RAS signaling epigenetically silences ERBIN expression in a reversible manner. Additionally, RASGRP1 and VPS28 emerged as key contributors to TMZ resistance by enhancing RAS-GTP transition and TMZ efflux, respectively.

Biosystems-derived exosomes (Exos) are composed of amphiphilic lipid bilayers encasing an aqueous core, providing a versatile platform for co-loading drugs with diverse physicochemical properties. A targeted Exos-based quadruple combination therapy significantly reduced tumor burden in vivo, suggesting a promising strategy for addressing tumor progression and TMZ resistance in NF1-mutated GBM [27,94].

Cyclin-dependent kinase (CDK) 7, a key component of transcription factor TFIIH, facilitates transcription initiation by phosphorylating serine-5 residues in the C-terminal domain (CTD) of RNA polymerase II (Pol II). Additionally, it functions as the catalytic subunit of the CDK-activating kinase (CAK) complex, promoting cell-cycle progression. CDK9, part of the P-TEFb complex, phosphorylates serine-2 residues on the Pol II CTD, enabling productive elongation and mRNA synthesis. Both kinases are crucial for transcription and cell division, since their inhibition suppresses transcription and translation, impairing the growth of actively dividing GBM cells while sparing post-mitotic neurons. CDK7 inhibition disrupts SE-associated transcription, which is essential for maintaining tumor cell growth and stemness. Similarly, CDK9 inhibitors target Pol II and p70S6K-dependent pathways, reducing the viability, invasiveness, and self-renewal of GBM, regardless of TMZ sensitivity.

Concurrently targeting multiple pathways might enhance GBM treatment by reducing compensatory pathway activation. CDK7 and CDK9 inhibitors exemplify this strategy and have demonstrated efficacy in hematological malignancies, showing potential for GBM therapy. However, their low BBB permeability constitutes a significant limitation [95].

Chromatin structural regulators, such as CCCTC-binding factor (CTCF) and Yin Yang 1 (YY1), have emerged as critical players in cancer biology. In gliomas, IDH1 mutations disrupt chromosomal neighborhoods through CTCF hypermethylation, while YY1 mediates chromatin loops and transcription elongation. YY1 was identified as a selective dependency in GSCs, regulating transcriptional CDK9 and RNA processing programs, including RNA splicing (SRSF1-3) and N6-methyladenosine (m6A) modification (METTL3 and YTHDF2). Targeting the YY1-CDK9 complex or associated transcription elongation machinery disrupts the self-renewal and proliferation of GSCs and induces interferon responses. This mechanism presents a novel therapeutic strategy, particularly in combination with immunotherapy, to address GBM’s resistance to treatment and limited immunotherapy efficacy [96].

TRIM24, a histone reader that binds to specific histone post-translational modifications, is highly expressed in GBM and drives GSC self-renewal and invasion while also functioning as a STAT3 transcriptional co-activator. Harvey Rat Sarcoma Viral Oncogene Homolog (HRas) is a member of the RAS protein family and is frequently activated in GBM, where the HRasV12 mutation activates the phosphorylated adaptor for RNA export (PHAX), which upregulates U3 small nucleolar RNAs (U3 snoRNAs) and recruits the Ku-dependent DNA-dependent protein kinase catalytic subunit (DNA-PKcs). TRIM24, when overexpressed, is also recruited by PHAX to U3 snoRNAs, where it facilitates DNA-PKcs phosphorylation of TRIM24 and induces reprogramming of the epigenome and transcription factor network, promoting the transformation into epithelioid GBM-like tumors. Additionally, targeting DNA-PKcs with the inhibitor NU7441 in combination with TMZ significantly enhances treatment efficacy for these tumors [97].

TRIM37, another member of the TRIM protein family, interacts with epigenetic complexes to regulate gene methylation and ubiquitination, influencing tumor proliferation and differentiation. In GSCs, TRIM37 is implicated in the activation of the SHH pathway, which regulates cell differentiation and contributes to GSC viability and invasion. TRIM37 interacts with EZH2 to epigenetically silence the SHH inhibitor PTCH1, leading to abnormal SHH pathway activation. TRIM37 knockdown reduces GSC stemness, promotes apoptosis, and extends survival in mouse models, constituting a potential therapeutic target for GBM, particularly targeting GSCs [98].

CELF2, part of the CELF/Bruno-like family of RNA-binding proteins, promotes a proliferative and OLIG2-positive phenotype in GSCs. CELF2 acts as a master epigenetic regulator by influencing the chromatin landscape through modulation of H3K9me3 levels via TRIM28 and G9a expression. This regulation represses genes like SOX3, which oppose the mitotic and OLIG2-positive tumor phenotype. These insights highlight CELF2’s role in GBM malignancy and its potential as a therapeutic target for combating the aggressive GSC phenotype [99].

Bromodomain and extra-terminal tail (BET) can regulate gene transcription by linking the histone code to transcriptional machinery, binding to acetylated chromatin residues to facilitate gene transcription. Small-molecule BET inhibitors (BETis) can disrupt these processes and modulate DNA damage response pathways, notably reducing MGMT expression by diminishing bromodomain-containing protein (BRD) 4 and Pol II binding at the MGMT promoter. By reducing MGMT expression, BETi sensitizes GBM cells to TMZ without compromising the mismatch repair system essential for alkylating agent efficacy. This suggests that incorporating BETi into the current standard of care with TMZ could sensitize GBM patients with unmethylated MGMT promoter [100].

Among BET family members, BRD4 plays a notable role in glioma progression and malignancy, particularly in R132H IDH mutants, where MYC activation is associated with transformation into higher-grade GBMs. Additionally, BRD4 expression is associated with reduced survival in IDH-mutant gliomas, while BRD3 expression is linked to poorer outcomes in IDH-wildtype gliomas. BETi JQ1 demonstrated significant inhibition of BET activity but limited clinical success [101], as seen with the withdrawal of the Phase II trial for birabresib (MK-8628, OTX015), a selective inhibitor of BRD2/BRD3/BRD4, due to lack of efficacy in IDH-wildtype gliomas. However, the study’s findings point to the potential of BETis, particularly those targeting BRD4, as promising therapeutics for IDH-mutant gliomas [102,103]. The clinical and in vitro evidence supports the hypothesis that IDH-mutant gliomas exhibit a preferential reliance on BET protein activity, offering a rationale for repurposing BETis to enhance therapeutic strategies for this glioma subtype [101].

Another BRD protein linked to gliomagenesis is BRD8, especially in TP53 wild-type GBM. BRD8 suppresses p53 activity by compacting chromatin via its bromodomain, sequestering the H2AZ histone variant. Targeting BRD8’s bromodomain remodels chromatin, restores p53 function, and selectively inhibits TP53 wild-type GBM growth. Thus, it constitutes a promising therapeutic target, with potential synergy when combined with MDM2 inhibition [104].

Phosphocreatine (PCr) is a metabolite essential for energy metabolism, primarily found in high-energy-demand tissues like neurons and muscles, and results from reversible transfer of phosphate from ATP to creatine by creatine kinases (CKs). Elevated PCr production is a distinct metabolic feature of GSCs, since PCr prevents the degradation of chromatin regulator BRD2, promoting proper chromosome segregation. The ZEB1-KAT2B complex promotes CKB (CK, brain-type) expression, increasing PCr levels in GSCs compared to differentiated tumor cells, emphasizing metabolic heterogeneity within GBM. Hypoxia further induces CKB expression via HIF-2α, supporting GSC survival in low-oxygen environments. Additionally, PCr metabolism may contribute to therapy resistance, considering its neuroprotective effects, such as shielding the brain from radiation damage. These findings underscore the potential of targeting PCr metabolism as a therapeutic strategy, addressing GSC-mediated resistance [105].

In GBM, chitinase 3-like 1 (Chi3l1) serves as a crucial marker for identifying the mesenchymal subtype and is linked to increased expression within oncogenic pathways, including NF-κB RelB and STAT-3/RTVP-1 signaling. Chi3l1 is a paracrine modulator of GSC states, promoting mesenchymal transcriptomic profiles while reducing the likelihood of transitioning to terminal states. Chi3l1 also induces chromatin remodeling, enhancing promoter accessibility enriched with Myc-associated zinc finger protein (MAZ) motifs and increasing MAZ transcription factor activity. Targeting Chi3l1 with therapeutic antibodies is a promising strategy to reduce tumor burden in GBM [106].

NF-κB signaling pathways and inflammatory response are also disrupted by Caspase-8 downregulation in GBM cells. Src-dependent phosphorylation of Caspase-8 at Y380 is a critical mechanism sustaining angiogenesis and promoting resistance to ionizing radiation. This Src/Caspase-8/NF-κB interplay suggests that Src kinase inhibitors like dasatinib could partake in GBM’s treatment, especially in mesenchymal subtypes [107].

KDM4B, a histone demethylase, influences cell proliferation, migration, and invasion in GBM while inducing G2/M cell cycle arrest. The oncogenic role of KDM4B is mediated through its JmjC domain, which catalyzes H3K9me3 demethylation, thereby activating transcription of downstream genes. KDM4B stabilizes MYC expression through epigenetic regulation, driving tumor progression. MYC is a well-known oncogene that promotes proliferation, metabolic reprogramming, and chemotherapy resistance in GBM, making KDM4B a valuable therapeutic target, particularly in cases with high MYC amplification [108].

KDM4C, another histone demethylase, is essential for GBM cell proliferation and positively correlates with c-MYC expression. KDM4C directly induces c-MYC expression and interacts with p53, demethylating its lysine 372, leading to decreased p53 stability and compromised apoptotic function. Targeting KDM4C could be a promising therapeutic approach for GBM [109].

Protein arginine N-methyltransferase 6 (PRMT6) promotes the transcription of CDC20, a cell cycle mediator, through histone arginine methylation (H3R2me2a). Elevated CDC20 levels facilitate the ubiquitination and degradation of CDKN1B, a G1/S phase cell cycle inhibitor, driving GBM cell proliferation and tumor progression. A small molecule inhibitor of PRMT6 was shown to suppress GBM cell proliferation in vitro, suggesting therapeutic potential of targeting the PRMT6-CDC20 axis [110]. Moreover, inhibition of PRMT6 with the small molecule EPZ enhances GSC sensitivity to radiotherapy, improving survival in preclinical models via CK2α-PRMT6-RCC1 signaling axis blockage. CK2α phosphorylates PRMT6, stabilizing it and enabling PRMT6 to methylate RCC1, which is necessary for chromatin binding and mitotic progression. As such, targeting PRMT6 and CK2α could improve GBM treatment outcomes by disrupting mitotic processes and overcoming resistance to therapy [111].

Methylthioadenosine phosphorylase (MTAP) deficiency in tumor cells reduces the expression of Fanconi anemia (FA) genes, critical for repairing interstrand cross-link (ICL)-induced DNA damage. PRMT5, an epigenetic regulator of arginine methylation overexpressed in GBM and linked to poor survival, maintains FA gene transcription partly by controlling H3R2me1 levels at their promoters. Blocking PRMT5 with small-molecule inhibitors replicated the effects of MTAP loss, showing reduced H3R2me1, diminished FA gene expression, and increased sensitivity to ICL agents. These results highlight an epigenetic mechanism linking MTAP deficiency to compromised DNA damage response pathways, identify vulnerability in MTAP-deficient cancer cells, and support the development of PRMT5-targeted therapies. Furthermore, the brain-penetrant PRMT5 inhibitor LLY-283 demonstrated significant efficacy in preclinical models, extending survival and targeting therapy-resistant GSCs. PRMT5 inhibitors show promise for improving outcomes in GBM, especially in patients unresponsive to TMZ [112,113].

SEs are clusters of enhancers enriched with transcription factors, chromatin regulators, co-activators, Pol II, and enhancer-associated marks like H3K27ac, playing a pivotal role in tumorigenesis [114]. Core stem cell pathways, such as WNT, NOTCH, and hedgehog, sustain GSC properties and inhibit apoptosis. Inhibition of epigenetic regulators like BRD4 suppresses hedgehog pathway genes and curbs tumor growth, especially in cases resistant to Smoothened antagonists.

KLHDC8A, regulated by SE and SOX2, is upregulated in GSCs, promoting hedgehog signaling through enhanced ciliogenesis. Primary cilia, crucial hubs for signaling pathways like hedgehog, WNT, and NOTCH, are present in 20–25% of GSCs and are tightly controlled during the cell cycle. Aurora B/C kinases have been identified as therapeutic targets, with GSCs showing heightened sensitivity to their inhibition compared to differentiated cells. Combining Aurora B/C kinase inhibitors with Smoothened inhibitors effectively disrupted GSC survival by targeting both ciliated and mitotic tumor cells. This dual approach highlights a novel therapeutic avenue for GBM treatment [115].

Rovalpituzumab tesirine (Rova-T), a Delta-like protein 3 (DLL3)-targeting antibody-drug conjugate, was evaluated for its efficacy in GBM as part of a broader phase I/II study on DLL3-positive tumors. DLL3, a ligand in the NOTCH signaling pathway, is implicated in tumor progression and is expressed in various tumor types, including GBM. For GBM, Rova-T demonstrated limited antitumor activity and significant toxicities, consistent with findings across other solid tumors. These results underscore the challenges of treating heavily pretreated refractory cancers, including GBM, and highlight the need for further investigation into DLL3 as a therapeutic target in GBM and other tumor types [116].

Histone H3 lysine 4 trimethylation (H3K4me3) marks regulatory elements of active or poised genes and influences GSC responses to hypoxia. DPY30, a component of the MLL complex, is a key regulator of H3K4me3 catalysis, with its knockdown leading to global H3K4me3 reduction. Targeting DPY30 selectively inhibited tumor formation in vivo without impairing GSC proliferation in vitro, suggesting its role in reprogramming the H3K4me3 landscape to support angiogenesis and hypoxia-related signaling in the tumor microenvironment (TME). This highlights DPY30 as a critical regulator of epigenetic modifications supporting GSC-driven tumor progression and identifies both DPY30 and PDE4B, a downstream target of DPY30, as potential therapeutic targets in GBM, with the PDE4 inhibitor rolipram as a promising agent [117].

### 4.4. Non-Coding RNA Targeting

Altered collagen signaling, especially the collagen VI (COL VI) family, has been implicated in the development and progression of CNS tumors [118]. COL VI components are also linked to poor prognosis and influence responses to anti-VEGF therapy [119]. miR-3189-3p regulates EMT and migration in GBM via H3K27me3 histone modifications, targeting COL6A2, a key collagen VI member. The EZH2/PRC2|miR-3189|COL6A2 axis drives EMT and tumor progression, highlighting its potential as a therapeutic target in GBM [40].

EZH2-mediated epigenetic modifications also lead to miR-490-3p downregulation in GBM. Higher miR-490-3p expression and its host gene, CHRM2, were correlated with improved patient survival, emphasizing its prognostic value. TGF-β signaling is a well-established regulator of the migratory phenotype in GBM cells. Functionally, miR-490-3p acts as a tumor suppressor, inhibiting migration and EMT by directly targeting the upstream receptor TGFBR1 and the downstream transcription factor TGIF2 within the TGF-β signaling pathway. This oncogenic role of the EZH2|miR-490-3p|TGIF2|TGF-β axis highlights its potential as a therapeutic target [41].

Progesterone has been shown to enhance cell proliferation, migration, and invasion in human GBM-derived cells by activating the progesterone receptor and regulating the expression of key genes involved in these processes, including TGF-β, COF1, EGFR, VEGF, and cyclin D1. Progesterone upregulates 8 miRNAs and downregulates 8 miRNAs, and its effects were blocked by the progesterone receptor antagonist RU486. Bioinformatic analyses suggested that progesterone regulates key processes in GBM, such as proliferation, cell cycle progression, and migration, through its modulation of miRNA-mRNA networks [120].

Reduced miR-146a expression is linked to shorter overall survival, independent of MGMT methylation status. Promoter methylation-induced miR-146a silencing contributes to tumor progression and resistance to therapy. Mechanistically, miR-146a inhibits GBM cell stemness by directly targeting POU3F2 and SMARCA5, two transcription factors that reciprocally regulate each other and whose expression levels are positively correlated in GBM. These findings underscore the potential of miR-146a, POU3F2, and SMARCA5 as promising therapeutic targets for GBM [121].

miR-219-1, a markedly downregulated miRNA in various cancers, produces two mature forms, miR-219-5p and miR-219-1-3p, each with distinct mRNA targets. In GBM, both forms exhibit reduced expression compared to normal tissues due to methylation sensitivity. Reactivating their expression with demethylating agents effectively lowered the levels of oncogenic target mRNAs and proteins, highlighting their potential as promising therapeutic targets [122].

The NF-κB pathway is triggered when cytokines stimulate the phosphorylation of inhibitory κB (IκB), leading to the degradation of IκBα and the subsequent release of NF-κB, which is then translocated to the nucleus and activates various genes. Chemotherapy and radiotherapy further activate NF-κB, contributing to tumor progression and poor prognosis. A crucial factor for maintaining NF-κB activation is miR-194-3p suppression, which influences pro-neural to a more aggressive and adaptable mesenchymal transition. The epigenetic regulation of NF-κB and its role in subtype plasticity show potential to reduce GBM heterogeneity and improve therapeutic outcomes [123]. The lncRNA XTP6 is upregulated in GBM and correlates with poor patient prognosis, since XTP6 facilitates the activation of the NF-κB pathway by downregulating IκBα and creating a positive feedback loop with transcription factor c-Myc. Indeed, the c-Myc/XTP6/NF-κB loop plays a key role in GBM malignancy and may be an interesting therapeutic target [124].

In GBM, the regulation of ferroptosis, a form of programmed cell death driven by iron-dependent lipid peroxidation, involves key epigenetic mechanisms, particularly the m6A RNA modification governed by the methyltransferase-like 3 (METTL3)/methyltransferase-like 14 (METTL14) complex, which modifies transcripts of genes such as glutathione peroxidase 4 (GPX4). Complement C5a receptor 1 (C5aR1), highly expressed in GBM cells, protects against ferroptosis by activating the ERK1/2 signaling pathway, upregulating METTL3, and stabilizing the m6A modification of GPX4. This promotes GPX4 expression and contributes to GBM progression. Targeting C5aR1 may disrupt the METTL3-dependent m6A modification of GPX4, offering a promising gene therapy approach to induce ferroptosis and inhibit GBM growth [125]. Indeed, RNA m6A methylation levels are reduced in GBM, with a notable decline in the m6A/A ratio in cells treated with TGF-β1. Decreased RNA m6A methylation promotes EMT and vasculogenic mimicry processes in GBM, with ALKBH5 upregulation or METTL3 downregulation driving increased tumor invasiveness [126].

Ribose 2′-O-methylation, the most prevalent modification in human ribosomal RNA (rRNA), plays a crucial role in ribosome biogenesis and the translational regulation of oncogenic proteins. This modification, mediated by C/D box small nucleolar ribonucleoproteins (snoRNPs), supports cell proliferation and survival. INHEG, a GSC-specific lncRNA, facilitates the interaction between the SUMO2 E3 ligase TAF15 and NOP58, a key component of snoRNP complexes responsible for guiding rRNA methylation. This interaction promotes NOP58 sumoylation, which in turn accelerates the assembly of C/D box snoRNPs, enhancing their function in rRNA methylation and supporting GSC self-renewal and tumorigenesis. This axis offers promising therapeutic avenues for targeting GSC [127].

Using multiomics techniques and molecular assays, a posttranscriptional regulatory circuit centered on the lncRNA DARS1-AS1 and its associated RNA-binding protein, YBX1, was identified. DARS1-AS1/YBX1 stabilizes mRNAs of regulators such as E2F1 and CCND1, forming a transcriptional/posttranscriptional feed-forward loop that promotes GSCs’ transition from G1 to S phase. DARS1-AS1 is also linked to enhanced DNA repair through the HR pathway. Silencing DARS1-AS1 sensitized GBM cells to radiation, suggesting that targeting the DARS1-AS1/YBX1 axis could impair HR, making GBM cells susceptible to radiation and poly-ADP-ribose polymerase (PARP) inhibitors. These findings propose a therapeutic strategy combining small interfering RNA (siRNA)/miRNA delivery systems targeting this axis with radiation or PARP inhibitors, warranting further exploration [128].

Amplification of the PDGFRA gene locus is a common event in GBM, promoting tumor development and progression. lncRNA LINC02283 is co-amplified with the PDGFRA locus and highly expressed in PDGFRA-driven high-grade gliomas, enhancing GBM malignancy by modulating PDGFRA and its downstream signaling pathways. Since clinical trials targeting PDGFRA with kinase inhibitors have been largely unsuccessful, therapeutic strategies involving antisense oligonucleotides or decoy RNAs to disrupt the LINC02283–PDGFRA interaction could be an approach to combat GBM malignancy [129].

Malate dehydrogenase 2 (MDH2) plays an important role in GBM metabolism, as well as the MDH2-interacting lncRNA malate dehydrogenase degradation helper (MDHDH). MDHDH directly binds to MDH2, promoting its degradation via interaction with PSMA1, which facilitates the proteasomal degradation of ubiquitinated MDH2. This process alters the NAD^+^/NADH ratio, inhibits glycolysis, activates the AMPK/mTOR pathway, and induces autophagy and apoptosis. Epigenetically, MDHDH expression is suppressed by the PRC2/EZH2 complex, with the PRC2 inhibitor GSK126 having therapeutic potential. Clinically, elevated MDHDH expression is associated with lower glioma WHO grades and improved patient survival, underscoring its potential as a biomarker and therapeutic target [130].

The mini-chromosome maintenance protein (MCM) gene family is critical for regulating cell cycle and mitigating DNA replication stress. MCM4 is highly expressed in GBM, significantly influencing cell proliferation and the cell cycle. Its expression is intimately related to a competing endogenous RNA (ceRNA) network, showing that lncRNAs and miRNAs regulate MCM expression and contribute to glioma progression. Therefore, MCM4 has been identified as a potential therapeutic target and prognostic marker in GBM, with reported associations to immune regulation, tumor progression, and drug sensitivity [131].

The lncRNA TCONS_00004099 is highly expressed in glioma, and its silencing decreases cell viability, migration, and invasion and promotes apoptosis through the generation of miRNAs. One TCONS_00004099-derived miRNA enhanced cell viability and reduced apoptosis by targeting PTPRF mRNA, inhibiting its translation, and promoting mRNA degradation, constituting a potential therapeutic target [43].

The association between SEs and lncRNAs has culminated in the development of a risk signature that functions as an independent prognostic indicator for glioma patients [114]. In IDH-wildtype GBM, a novel SE-associated lncRNA, cancer stem cell-associated distal enhancer of SOX2 (CASCADES), has been identified as a key regulator of GSC identity through the epigenetic regulation of SOX2 [29]. LINC00945 is another SE-lncRNA that promotes glioma cell proliferation, EMT, migration, invasion, and tumor growth in xenograft models, constituting a novel therapeutic target. Additionally, SE-lncRNAs such as TMEM44-AS1, CCAT1, LINC00152, and NEAT1 are identified as facilitators of glioma malignancy, reinforcing the prognostic and therapeutic potential of SE-lncRNA signatures [114].

### 4.5. Epigenome Editing

Efforts to overcome TMZ chemoresistance in GBM have shown limited success in clinical trials. Direct MGMT inhibition using O-6-benzylguanine faced feasibility issues due to severe toxicities in a phase I trial [132], while dose-dense TMZ strategies failed to improve outcomes in a phase III clinical trial [49]. One strategy to address these challenges was the design of a clustered regularly interspaced short palindromic repeats (CRISPR)-based approach targeting MGMT with a dCas9-DNMT3A catalytic domain fusion protein (d3A). This system enabled site-specific DNA methylation without altering the gene sequence, leveraging single guide RNA (sgRNA) specificity and reversible epigenetic modifications. Using multiple sgRNAs, comprehensive methylation was achieved, leading to MGMT downregulation and increased TMZ sensitivity in vitro. Genome-wide and transcriptome-wide analyses confirmed minimal off-target effects, highlighting the potential of d3A/CRISPR-directed methylation as a safe and effective therapeutic strategy to enhance TMZ chemosensitivity in GBM [52].

CRISPR-based genomic perturbations have also been applied to novel EGFR enhancer elements near the GBM-associated single nucleotide polymorphism rs723527. Targeting these enhancer regions reduces proliferation and migration, partly by increasing apoptosis, possibly due to metabolic reprogramming. This approach diminishes the malignancy of GBM cells, making them more sensitive to TMZ. Combining such genomic interventions with existing treatments may improve therapeutic outcomes [133].

The need for rapid identification of novel cancer-related mechanisms and the development of effective therapies has driven the pursuit of robust and affordable screening methods able to detect epigenetic factors that influence tumor survival. One example is the Epigenetic Domain-specific Knock Out Library (EpiDoKOL), a customized tool designed to target the functional domains of key epigenetic modifiers. Through drop-out screens in various cell lines, it identified ASH2L as a critical gene for GBM cell survival. ASH2L, a member of the trithorax group family, is involved in methyltransferase complexes that regulate H3K4 methylation, which supports active gene transcription. Elevated WDR82 and H3K4me3 levels have been linked to therapeutic sensitivity in GBM. ASH2L directly influences the regulation of cell cycle genes and promotes tumor cell survival both in vitro and in vivo. These findings highlight the potential of chromatin-focused CRISPR library screens, like EpiDoKOL, for identifying novel epigenetic vulnerabilities in GBM [134,135].

### 4.6. Combination Therapies

Vorinostat, FDA-approved for cutaneous T cell lymphoma, has shown modest effectiveness as a standalone therapy in recurrent GBM [136]. However, its limited success in treating solid tumors has encouraged exploration of synergistic strategies. PARP inhibitors like olaparib have shown promise in targeting DNA repair pathways in GBM, particularly in cells with compromised HR [137]. Based on vorinostat’s ability to suppress HR gene expression, its combination with olaparib significantly improved the elimination of GBM cells by reducing DNA repair capabilities and inducing apoptosis. This dual treatment disrupted the cell cycle, causing G2/M phase arrest, increased DSBs, and oxidative DNA damage. The combined therapy demonstrated efficacy across multiple GBM models, with both drugs showing minimal toxicity to healthy tissues [138].

Phospholipase D (PLD), especially its isoforms PLD1 and PLD2, is implicated in tumor malignancy, GSC maintenance, and resistance to therapies. Furthermore, vorinostat induces PLD1 upregulation, which protects GBM cells from apoptosis and contributes to resistance. Inhibition of PLD1, when combined with vorinostat, significantly reduces GBM invasiveness, angiogenesis, self-renewal, and intracranial tumor formation, effectively overcoming resistance to conventional therapies. Considering that GSCs drive therapy resistance, targeting PLD1 along with vorinostat is a promising strategy. Likewise, developing biomarkers to predict therapeutic efficacy could facilitate the precise selection of candidates for such innovative combination treatments [139].

To overcome the poor BBB penetration and limited GBM targeting of BRD4 inhibitor OTX015, a cell membrane coating approach was attempted, co-encapsulating it with TMZ. The resulting nanomedicine, ABNM@TMZ/OTX, disclosed strong synergy between TMZ and OTX, effective BBB penetration, GBM targeting, and immune response activation, with safety profiles supporting its clinical application [140].

Lysine-specific histone demethylase 1 (LSD1), an epigenetic enzyme overexpressed in various malignancies, and HOTAIR, an lncRNA associated with GBM prognosis, are compelling targets for clinical translation. Although existing EZH2 inhibitors, like tazemetostat, show limited efficacy in certain hematological malignancies, and inhibitors like JQ1 effectively target HOTAIR transcription, the complexity of epigenetic regulation underscores the limitations of single-drug therapies, including off-target effects and global impacts. The combination of the HOTAIR-EZH2 disruptor AQB and the LSD1 inhibitor GSK-LSD1 induced cell cycle arrest and apoptosis in GBM cell lines, demonstrating greater antitumor efficacy than either agent alone. The combination therapy approach is particularly advantageous, as both AQB and GSK-LSD1 are readily synthesizable and commercially available. The synergistic effects of combining agents can allow for reduced dosages and better-targeted tumor suppression. Additionally, the possibility of integrating AQB and GSK-LSD1 with other epigenetic modifiers into triple or quadruple regimens opens new avenues for GBM therapy [141,142].

Combinations of histone methylation and acetylation inhibitors, such as BIX01294 (G9a inhibitor), DZNep (EZH2 inhibitor), TSA (HDAC inhibitor), and RG-108 (DNMT inhibitor, DNMTi), hold promise for targeting GBM cells while sparing normal stem cells. Initial experiments identified effective concentrations of these inhibitors, with medium doses of BIX01294 and TSA significantly reducing GBM cell viability while causing minimal harm to human mesenchymal stem cells. In contrast, DZNep affected both cell types similarly, and RG-108 showed no significant effect. Combining low and medium concentrations of these inhibitors demonstrated synergistic effects, selectively enhancing GBM cell death while preserving normal stem cells. Notably, TSA and BIX01294 at medium concentrations were particularly effective, offering a refined therapeutic strategy for targeting GBM with minimal off-target effects [143].

Additionally, DZNep was combined with panobinostat (HDACi) and evaluated for synergistic effects along with TMZ and APR-246, which restores p53 function in mutated GBM. DZNep and panobinostat together exhibited the strongest synergistic effects against GBM cells, significantly enhancing apoptosis and reducing clonogenicity. The combination of panobinostat and TMZ showed moderate synergy, while DZNep and TMZ displayed slight synergy. When APR-246 was added to these combinations, its effects were primarily additive rather than synergistic [144].

Estrogen receptor β (ERβ) functions as a tumor suppressor, and its expression is often reduced during cancer progression, including in gliomas. HDACis, such as panobinostat and romidepsin, increase ERβ expression and enhance its signaling. This upregulation sensitizes GBM cells to ERβ agonist therapy, leading to reduced cell viability and invasiveness and increased apoptosis. In mouse models, the combination of HDACi and ERβ agonist improved survival, suggesting that HDACi combined with ERβ agonists could be a novel therapeutic strategy for GBM [145].

The PI3K pathway, a critical driver of tumor growth and survival, is frequently activated in GBM through mutations in genes like PTEN and EGFR. Despite advancements in PI3K-targeted therapies, including dual PI3K/mTOR inhibitors (e.g., NVP-BEZ235), resistance often emerges due to induction of stem cell-associated genes, potentially promoting recurrence. These stem genes are commonly activated via the JAK/STAT3 pathway, which is itself constitutively active in GBM due to mutations in JAK1/2 and EGFR. While JAK inhibitors, such as ruxolitinib and AZD1480, have demonstrated efficacy in reducing tumorigenesis and invasiveness in preclinical studies, their effectiveness as monotherapies is limited. Combining PI3K inhibition with JAK inhibitors was hypothesized to suppress both tumor growth and stem gene expression, potentially improving therapeutic outcomes. However, studies have shown limited efficacy of this combination, likely due to signaling redundancies that sustain stem characteristics [146].

### 4.7. Immunomodulation via Epigenetics

Immunotherapy shows promise in treating various cancers but faces significant challenges in GBM. These obstacles include limited immune cell infiltration, an immunosuppressive microenvironment, and poor recognition of tumor-associated antigens by immune effector cells. A promising strategy involves developing cancer vaccines based on tumor-specific antigens. These antigens often arise from noncanonical peptides generated by cancer-specific mutations, known as neoantigens, which are presented on human leukocyte antigen class I (HLA-I) molecules. Although GBM vaccines targeting neoantigens have shown immunogenic potential in clinical trials, their effectiveness is limited by GBM’s low mutational burden and antigen load and is often specific to individual patients. Moreover, cancer’s ability to evade immunosurveillance through immunoediting makes single-target vaccines universally ineffective [147,148,149].

Cancer-testis antigens (CTAs) are proteins normally expressed in germinal spermatogonia but silenced in somatic tissues through DNA methylation and other epigenetic mechanisms. In many cancers, including GBM, CTAs like New York esophageal squamous cell carcinoma 1 (NY-ESO-1) are aberrantly re-expressed. Prior studies have shown that decitabine, a DNMTi, can induce CTA expression in GBM without affecting normal tissue [150]. Because in GBM NY-ESO-1 expression is silenced by CpG hypermethylation, decitabine reactivates NY-ESO-1 expression by reversing DNA hypermethylation, enhancing CD8^+^ T cell response, and leading to robust antitumor activity. Furthermore, decitabine upregulates other CTAs across glioma populations, reactivates human endogenous retroviruses (HERVs), and boosts an antiviral-like interferon signaling response, further enhancing immunogenicity. Transcriptomic profiling confirmed that decitabine-induced CTA expression could support polyclonal T cell and vaccine therapies, emphasizing its potential as an immunosensitizing agent [148].

HERVs, particularly HML-6 and its gene product ERVK3-1, have been linked to glioma progression, since HERV expression correlates with DNA hypomethylation at specific loci. Notably, HML-6 is overexpressed in aggressive GBM models, and elevated ERVK3-1 expression was associated with reduced survival in GBM patients [151].

Other favorable immunogenic antigens are transposable elements (TEs) and cryptic promoters, which may be reactivated by epigenetic drugs, such as decitabine and panobinostat. These activated TEs generate double-stranded RNAs, triggering antitumor interferon responses, and can splice into downstream genes to form TE-chimeric antigens presented as HLA-I antigens. Notably, epigenetic drugs might also activate TEs in normal primary cells (although less significantly in quiescent cells), which has important clinical implications and highlights the need for careful candidate selection. Nevertheless, reactivation of underappreciated TE-based antigens absent from current whole-exome sequencing pipelines offers a novel avenue for personalized cancer vaccine strategies, providing synergistic benefits along with chemotherapy, radiation, or immunotherapy [149].

Peripheral immune cell counts have been linked to tumor cell states and immune infiltration, being considered as biomarkers in GBM when integrated with DNA methylation-based subclassification. Elevated neutrophil counts are associated with worse overall survival in newly diagnosed GBM, while decreases in lymphocytes, monocytes, and platelets correlate with progression and poor outcomes in recurrent disease. These findings support a subclass-specific approach to immunotherapy in GBM and underscore the potential of peripheral blood profiling, when interpreted through an epigenetic lens, to improve prognostication and patient selection [152].

Macrophages play a critical role in GBM progression and response to therapy, making them attractive therapeutic targets, although universal macrophage-targeting approaches are complicated by GBM’s inherent heterogeneity and its associated TME. Tumor-associated macrophages (TAMs), which include brain-resident microglia and infiltrating monocyte-derived macrophages, represent the most abundant immune cells in GBM TME and can predict patient survival. These cells exhibit distinct metabolic profiles and spatial and functional heterogeneity, particularly in the context of recurrence following radiotherapy. Indeed, a metabolic interplay takes place between mesenchymal-like GBM cells and lipid-laden TAMs, establishing a dynamic relationship centered on the recycling of cholesterol-rich myelin debris. GBM cells exploit macrophages’ capacity for myelin processing (originally a neuronal homeostatic function), leading to lipid overload in TAMs. This lipid accumulation reprograms TAMs into a pro-tumorigenic and immunosuppressive phenotype through transcriptional changes, chromatin alterations, and lipidomic rewiring, fueling GBM cell survival and proliferation. This emphasizes the need for therapeutic strategies targeting the metabolic interactions between GBM cells and TAM subsets to disrupt this protumorigenic cycle and improve clinical outcomes [153].

Blocking integrin β1 on monocytic myeloid-derived suppressor cells reduces TAMs, while dipeptidyl peptidase-4 (DPP-4) inhibition also diminished macrophage abundance, suggesting these molecules as potential immunotherapy targets in GBM. Interestingly, DPP-4 inhibition also affected other immune populations, likely due to tumor shrinkage or DPP-4 expression on other immune cells, such as T cells. While DPP-4 inhibition did not directly induce tumor cell death or alter activation/exhaustion markers in tumor-infiltrating CD8^+^ T cells, it stimulated splenic T cell proliferation in vitro. Integrin β1 is associated with poor prognosis in GBM and is overexpressed in models of GBM resistant to antiangiogenic therapies. Its involvement in tumor cell proliferation and self-renewal highlights its potential as a therapeutic target, offering a dual benefit in combating GBM. However, challenges arise due to its widespread expression and critical roles in normal brain function. Furthermore, DPP-4 stands out as a viable therapeutic candidate. DPP-4 inhibitors, commonly used to treat type II diabetes, could be repurposed as anticancer agents to effectively modulate immune responses [4,154].

Molecularly altered pathways associated with programmed death-ligand 1 (PD-L1), a key target in personalized cancer therapy, have been identified in GBM. These suggest novel molecular targets, including miR-196B, for precision medicine. While high PD-L1 expression correlates with potential benefits from pembrolizumab therapy, challenges like the BBB persist. Developing small molecules targeting identified RNAs and genes could improve therapeutic efficacy, warranting further clinical studies [155].

KDM6B, a histone demethylase that promotes gene transcription by demethylating the repressive histone mark H3K27me3, is highly expressed in myeloid subsets present in the TME. Epigenomic analyses revealed that KDM6B directly regulates H3K27me3 at anti-inflammatory genes, which inhibit pro-inflammatory pathways like cytokine production and phagocytosis. Pharmacological inhibition of KDM6B mimicked these effects, promoting pro-inflammatory phenotypes and improving sensitivity to anti-PD1 therapy. The findings provide compelling evidence that targeting KDM6B-mediated epigenetic pathways can reprogram intratumoral myeloid cells into a pro-inflammatory state, overcoming immune suppression in the TME and enhancing immune checkpoint therapy responses [156].

Protein disulfide isomerase A5 (PDIA5) is significantly associated with immune cell infiltration, immune pathways, and other immune-related signatures. In GBM, PDIA5 interacts with immune cells, and PDIA5 silencing led to increased PD-L1 and SPP1 expression, reduced proliferation, colony formation, and invasion abilities, while impairing migratory capacity of cocultured M2 macrophages. Additionally, PDIA5 exhibited predictive potential for immunotherapy response, highlighting its relevance as a biomarker and therapeutic target for cancer immunotherapy [157].

Due to the high and specific expression of EGFR in GBM, EGFR CAR-T cells hold significant therapeutic potential. While these cells effectively suppressed GBM cell growth in vitro and tumorigenesis in vivo, resistance quickly emerged. This resistance was linked to upregulation of immunosuppressive genes, including inhibitory immune checkpoints and inflammatory cytokines, driven by EGFR CAR-T cell-induced active enhancers. BRD4 inhibition with JQ1 disrupted immunosuppressive gene activation, and its combination with EGFR CAR-T therapy reduced immunosuppression, effectively curbing tumor growth and metastasis in GBM xenografts [158].

### 4.8. Emerging Technologies

Mutations in IDH1/2 and PPM1D may silence the nicotinic acid phosphoribosyl transferase (NAPRT) gene, essential for NAD^+^ production via the Preiss-Handler salvage pathway. Alternatively, NAPRT silencing makes glioma cells reliant on the nicotinamide phosphoribosyl transferase (NAMPT)-dependent nicotinamide salvage pathway for NAD^+^ synthesis. This vulnerability enables synthetic lethality using NAMPT inhibitors (NAMPTis), such as GMX1778, making NAPRT silencing a critical biomarker for NAMPTi therapy. However, NAMPTis face challenges, including low brain bioavailability and dose-limiting toxicities. Instead, NP formulations of NAMPTis enable rapid cellular uptake, significant intracellular NAD^+^ depletion, and selective efficacy in GBM models while avoiding hematologic and retinal toxicities observed with systemic delivery. Intracranial GBM mouse models show reduced tumor burden and improved survival with NP-loaded NAMPTi monotherapy [39].

Since HDAC6 is overexpressed in GBM, associating with poorer survival, and monoamine oxidase A (MAO A) is also upregulated in GBM, dual targeting of these enzymes seems a promising strategy. HDAC-MB, a novel multifunctional small-molecule probe designed for glioma theranostics, combines HDAC6 imaging, MAO A inhibition, and photodynamic therapy functions. Initially, the probe remains inactive, but upon activation by overexpressed HDAC6 in glioma cells, it releases methylene blue, which inhibits MAO A and restores the probe’s near-infrared fluorescence and photodynamic therapy activity when exposed to near-infrared light. This enables a highly selective and sensitive tool for detecting HDAC6 levels while also inducing cell death through the generation of reactive oxygen species (ROS) upon light activation. Additionally, the probe’s MAO A inhibition enhances its ability to reduce glioma cell migration, invasion, and proliferation [159].

Hydrolyzed rutin, a modified flavonoid derived from rutin deglycosylation, showed potential as an epigenetic drug due to its antiproliferative effects related to cell cycle inhibition in human GBM cell lines, reducing GBM aggressiveness. Besides its potential as a therapeutic agent, its prophylactic treatment use is particularly noteworthy, as it has been proven to decrease the aggressive behavior of GBM during primary treatment and prevent relapse, a benefit not yet reported for other adjuvant therapies. Hydrolyzed rutin’s action may also alter the response to conventional therapies, making it crucial for both primary and recurrent glioma treatment. The compound inhibits tumor growth and reduces anaplasia without genotoxicity, suggesting epigenetic modulation as a mechanism. Although preclinical results are promising, further investigation in orthotopic models and clinical studies is required to confirm its therapeutic efficacy [160].

Suicide gene therapy employs genes encoding enzymes that convert non-toxic prodrugs into toxic metabolites, selectively eliminating targeted cells. A prominent example is the herpes simplex virus-thymidine kinase (HSV-TK)/ganciclovir (GCV) system, which produces a toxic metabolite, GCV triphosphate, through cellular phosphorylation. This metabolite diffuses to adjacent cells, inducing apoptosis via a bystander effect. The tumor necrosis factor-related apoptosis-inducing ligand (TRAIL) enhances the efficacy of HSV-TK/GCV by increasing target-specific killing and amplifying the bystander effect. TRAIL, naturally expressed in the immune system, induces apoptosis in glioma cells, making it a promising GBM treatment. However, its therapeutic use is limited by rapid systemic clearance and glioma cell resistance, requiring improved delivery systems and tumoricidal efficacy.

Polyethylenimine (PEI), a widely used gene delivery vector, offers high transfection efficiency but is hampered by cytotoxicity. Modifications such as conjugating PEI with polylysine (PLL) have shown promise in reducing toxicity and improving gene delivery efficiency. In a preclinical study, a PEI-PLL copolymer was used to transfect mesenchymal stem cells with HSV-TK and TRAIL genes, creating a dual-functional therapeutic system for glioma. These genetically modified stem cells converted GCV into its toxic metabolite while synergistically enhancing TRAIL-induced apoptosis. This approach demonstrated significant antitumor effects in vitro and in vivo, including reduced cell proliferation and angiogenesis and increased apoptosis in glioma models [161].

A novel multimodal therapy, PEG-AuNPs@Hyp, demonstrates significant anticancer potential against GBM by targeting epigenetic and mitochondrial pathways. This red-light-responsive nanoformulation exhibits controlled drug release, effective uptake in GBM models, and enhanced tumor ablation in vivo. PEG-AuNPs@Hyp suppresses PcG proteins (EZH1, EZH2, and Bmi-1) via ubiquitin-mediated degradation and disrupts the interaction between PcG proteins and mutant IDH2. This process restores tumor suppressor gene activity by reducing histone trimethylation and promoter methylation. In addition, it enhances apoptosis through ROS generation, caspase-3 activation, mitochondrial membrane disruption, and downregulation of anti-apoptotic markers like Bcl2 [162].

### 4.9. Future Directions and Emerging Therapeutic Approaches

Advancing the standard treatment for GBM requires translating preclinical findings into clinical trials. However, this faces significant challenges, including the time and resources required, difficulties in patient enrollment, and limited effectiveness of many new approaches, all of which hinder the development of broadly applicable treatments.

HDACis aim to modify chromatin structure and gene expression. A trial explored the maximum tolerated dose of belinostat combined with radiotherapy and TMZ while also assessing the utility of magnetic resonance spectroscopic imaging in predicting better outcomes and detecting early treatment response [75]. Another study investigated convection-enhanced delivery of MTX110, a panobinostat formulation, directly to the tumor during surgery. This approach bypassed the BBB and delivered high concentrations of the drug to the tumor site [163].

Valproic acid, traditionally used to treat seizure disorders, has been explored for its potential to enhance the effects of standard radiotherapy and TMZ in children and adolescents with high-grade gliomas, including GBM [164]. Additionally, an open-label phase II study evaluated the safety and efficacy of prolonged doxorubicin administration combined with radiotherapy, TMZ, and valproic acid in newly diagnosed GBM and diffuse intrinsic pontine glioma. This trial was ultimately terminated due to the high heterogeneity of the enrolled patient population [165].

A pilot study aimed to assess the safety and feasibility of combining nivolumab, an immunotherapeutic agent, with stereotactic radiosurgery (gamma knife therapy) and valproic acid in recurrent GBM patients. However, this study was terminated early when the pharmaceutical company ceased providing nivolumab [166]. Similarly, a phase I trial evaluated the combination of pembrolizumab, another immune checkpoint inhibitor, with vorinostat, TMZ, and radiotherapy for newly diagnosed GBM [167].

Recent clinical trials have shown promising results for BETi in GBM treatment. A phase Ib dose-escalation trial with trotabresib (CC-90010) combined with radio-chemotherapy and TMZ reported good tolerability. Additionally, ongoing studies aim to assess the drug’s BBB penetration and its ability to deplete MGMT and other target proteins in a multi-center, open-label study [168,169]. A phase IIa trial investigated birabresib in recurrent GBM following standard therapy failure, aiming to determine the maximum tolerated dose. This study was terminated due to lack of efficacy [103].

A phase 1/2 study investigated olutasidenib (FT-2102), a mutant IDH1 inhibitor, in patients with advanced solid tumors and gliomas, either as monotherapy or combined with azacitidine for glioma treatment. However, the study yielded limited success [170]. After significantly impeding tumor progression in both subcutaneous and intracranial patient-derived xenograft models, an ongoing clinical trial is focusing on the IDH1-R132 mutation, assessing the safety, tolerability, pharmacokinetics, pharmacodynamics, and antitumor efficacy of DS-1001b [171,172].

A phase 0 first-in-human study investigated the safety of NU-0129, a novel spherical nucleic acid gold nanoparticle therapy, in patients with recurrent GBM or gliosarcoma undergoing surgery. Designed to cross the BBB, NU-0129 delivers nucleic acids targeting the Bcl2L12 gene, which promotes tumor growth by preventing apoptosis. Among eight enrolled patients, only one experienced serious adverse events [173].

Other key clinical approaches include identifying novel biomarkers, such as microRNAs, to develop signatures discriminating GBM from other malignant CNS tumors and enhancing diagnostic accuracy [174]. Another study focuses on integrating stem cell analysis, multiomics (including immunomics), and artificial intelligence to advance personalized GBM care by identifying immune markers for prognosis, testing GBM stem cell sensitivity to treatments, and creating ethical guidelines for artificial intelligence-assisted predictions [175]. Moreover, circulating microRNAs are being studied as biomarkers for monitoring diffuse gliomas, with the goal of differentiating true tumor recurrences from false positives on MRI [176].

Collectively, these trials aim to evaluate the safety, feasibility, and efficacy of innovative epigenetic therapies for GBM management, with an emphasis on combination strategies to improve therapeutic outcomes. An overview of these trials is summarized in Table A3.

## 5. Conclusions

GBM remains one of the most challenging cancers to treat, with a dismal prognosis despite advancements in therapy. Epigenetics emerges as a promising but underutilized therapeutic approach, targeting aberrant DNA methylation, histone modifications, and dysregulated ncRNAs that drive tumor progression, immune evasion, and therapy resistance. Early studies highlighted the potential of DNMTis, HDACis, and RNA-based therapies, especially in combination with immune checkpoint or PARP inhibitors, disclosing synergistic effects in preclinical models.

Clinical translation, however, faces considerable challenges, including BBB limiting drug delivery, bioavailability issues, off-target effects, systemic toxicity, GBM tumor heterogeneity, and adaptive resistance mechanisms. Precision medicine approaches, leveraging biomarkers to tailor interventions, are crucial to overcoming these barriers.

This scoping review has some limitations that should be acknowledged. Like most scoping reviews, no critical appraisal of the included studies was conducted, which limits the ability to assess the strength or quality of the evidence. While data charting was verified, the initial screening and extraction were primarily performed by one author, introducing potential bias. Additionally, the review focused narrowly on therapeutic targeting, excluding broader, although eventually clinically relevant, aspects, such as diagnostic or prognostic epigenetic research.

Future progress will rely on interdisciplinary strategies, integrating molecular biology, nanotechnology, and multiomics to refine knowledge on epigenetic landscapes and therapeutic combinations. Collaborative efforts across research disciplines and clinical domains are essential to accelerate the translation of epigenetic discoveries into standard care. In conclusion, the therapeutic potential of targeting epigenetics in GBM is undeniable, offering a path to overcome some of the limitations of current treatments. Advances in drug delivery, target specificity, and overcoming resistance mechanisms, combined with interdisciplinary approaches, may enhance the effectiveness of epigenetic therapies in improving outcomes for GBM patients.

## Figures and Tables

**Figure 1 ijms-26-05634-f001:**
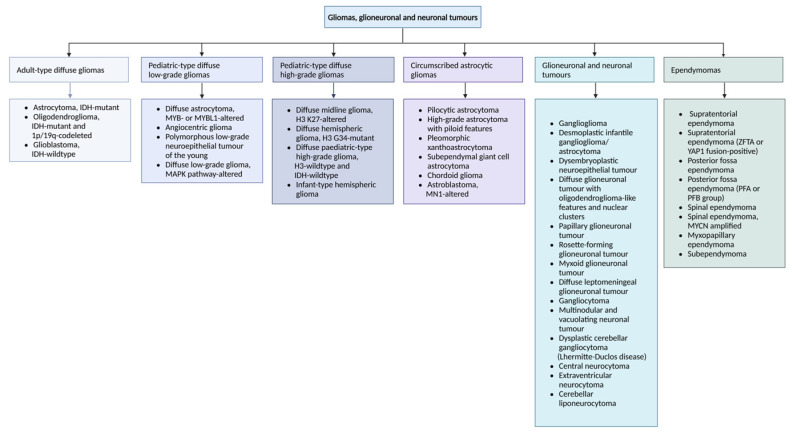
Overview of the World Health Organization 2021 central nervous system glial, glioneuronal, and neuronal tumor classification. Created in BioRender. Gonçalves Meleiro, M. (2025). https://BioRender.com/7tu8grq (accessed on 27 April 2025).

**Figure 2 ijms-26-05634-f002:**
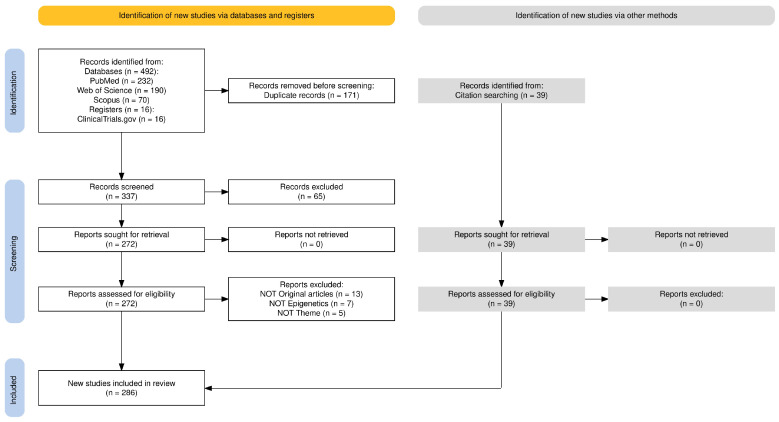
PRISMA-ScR flow diagram, according to “PRISMA 2020 flow diagram for new systematic reviews, which included searches of databases, registers, and other sources”.

**Figure 3 ijms-26-05634-f003:**
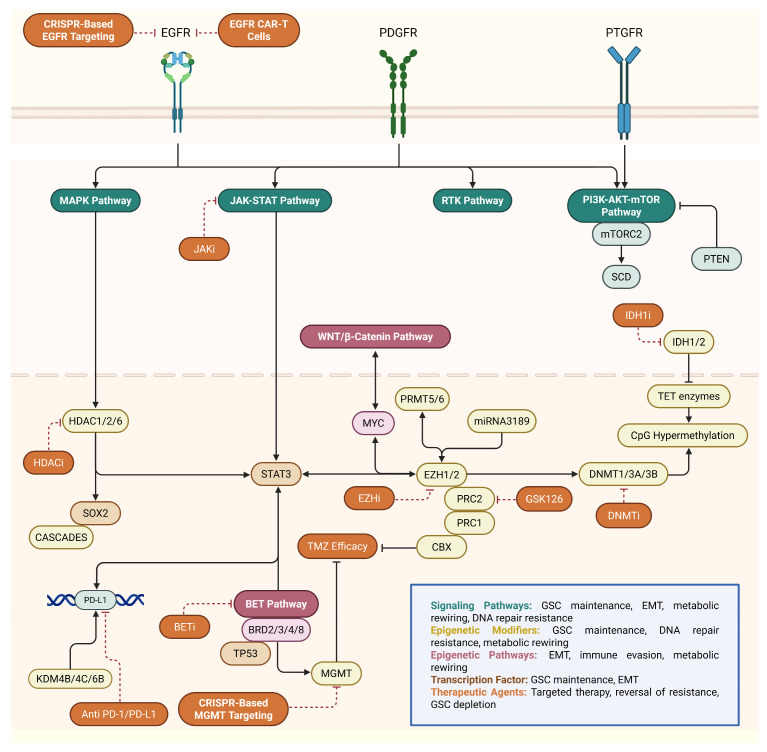
Simplified overview of epigenetic and signaling pathways in GBM pathogenesis and therapeutic targeting. Created in BioRender. Meleiro, M. (2025). https://BioRender.com/8e6i9hg (accessed on 26 May 2025).

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
