# Peer review of "Epigenetic Alterations in Glioblastoma Multiforme as Novel Therapeutic Targets: A Scoping Review"

_ijms, 2025, doi:10.3390/ijms26125634_

Round 1

Reviewer 1 Report

Comments and Suggestions for Authors

This scoping review provides a synthesis of recent findings on epigenetic mechanisms and their therapeutic potential in GBM. The manuscript is clearly structured and well-written, covering DNA methylation, histone modifications, and non-coding RNA regulation, as well as a wide range of epigenetic drugs and experimental therapies under preclinical and clinical evaluation.

Here you can find my comments

Major Comments

While the review is labeled as a “scoping review,” the methodology section would benefit from greater clarity and formal adherence to PRISMA-ScR guidelines. Please include:

    • A clear research question/objective aligned with the PCC (Population, Concept, Context) framework.
    • A supplementary table with full references for the included 273 studies, or at least an appendix summarizing main included works by category (DNA methylation, histone modification, ncRNA, etc.).
    • A critical synthesis of the quality of the studies would also enhance the manuscript, even if formal bias assessment was not performed.

Some sections (e.g., histone modification, EZH2, HDACs) are extremely dense and could benefit from condensation. Consider focusing more sharply on mechanistically distinct, translationally relevant findings and minimizing repetition across subsections. Several signaling axes are re-explained in multiple parts of the manuscript.

Given the complexity of the content, the manuscript would greatly benefit from:

    • A summary table of the major epigenetic targets, associated pathways, and therapeutic agents discussed and/or a graphical abstract or figure showing the integration of epigenetic modifications with therapeutic strategies in GBM.

Minor Suggestions

  • Revise some verbose or speculative statements, e.g., "holds the potential to redefine GBM treatment paradigms..." should be made more evidence-based.
  • Ensure consistency in terminology (e.g., GBM vs. glioblastoma).
  • To strengthen the review’s completeness, please consider citing the following article when discussing molecular predictors and in glioblastoma:
    • Saaid et al.: https://doi.org/10.3390/genes13081439.
      This paper contextualizes epigenetic and genetic predictors in GBM, including MGMT and IDH-related methylation profiles, and should be cited when discussing the predictive relevance of DNA methylation and IDH mutations (e.g., lines 88–95 or 264–266).
    • Along with citation 6 at line 99 I would cite this review (https://doi.org/10.3390/biomedicines11061520) regarding immunotherapy and oncolytic viruses in glioma
  • Define all abbreviations at first mention in each section (e.g., PRMT, GSC, SE).
  • Add page/figure/table numbers to improve navigation through the manuscript.

Author Response

3. Point-by-point response to Comments and Suggestions for Authors

Comment 1: This scoping review provides a synthesis of recent findings on epigenetic mechanisms and their therapeutic potential in GBM. The manuscript is clearly structured and well-written, covering DNA methylation, histone modifications, and non-coding RNA regulation, as well as a wide range of epigenetic drugs and experimental therapies under preclinical and clinical evaluation.

Response 1: We greatly appreciate your attention to detail and the helpful remarks provided throughout the review.

Comment 2: While the review is labeled as a “scoping review,” the methodology section would benefit from greater clarity and formal adherence to PRISMA-ScR guidelines.

Response 2: We thank the reviewer for this insightful comment. We would like to note that formal adherence to the PRISMA-ScR guidelines was already included in the protocol, as stated in the Methods section: “The protocol was designed according to the PRISMA Extension for Scoping Reviews (PRISMA-ScR) recommendations [15]” (page 4, lines 107–108). The complete PRISMA-ScR Checklist was submitted as non-publishing material. However, we agree that resubmitting it as a supplementary document (Supplementary Table 1) would improve clarity for the reader, and we have done so accordingly.

Comment 3: A clear research question/objective aligned with the PCC (Population, Concept, Context) framework.

Response 3: We appreciate this valuable suggestion and have revised the manuscript accordingly to clarify this point. The following sentence was added: “This scoping review was guided by the PCC framework, focusing on glioblastoma patients (Population), epigenetic alterations as therapeutic targets (Concept), and the limited effectiveness of current standard treatments (Context)” (page 4, lines 108-110).

Comment 4: A supplementary table with full references for the included 273 studies, or at least an appendix summarizing main included works by category (DNA methylation, histone modification, ncRNA, etc.).

Response 4: Thank you for your comment. This aspect was addressed in the original submission (Table 1, Table 2, and Table 3, pages 31–36), but we understand that it may have been overlooked. We believe that organizing the 273 included studies into three separate tables – “Epigenetic-related pathophysiology and potential therapeutic targets in GBM,” “Epigenetic-related molecules and drugs in GBM,” and “Most relevant clinical trials incorporating epigenetics in GBM” – offers a more accessible format while still presenting the full references.

Comment 5: A critical synthesis of the quality of the studies would also enhance the manuscript, even if formal bias assessment was not performed.

Response 5: We appreciate the reviewer’s suggestion. Given that this is a scoping review encompassing a substantial number of articles, it is not possible to perform a formal risk of bias assessment. Although a critical synthesis of the quality of the included studies was briefly addressed in the manuscript, we acknowledge that it may not have been sufficiently emphasized. We have now revised the relevant paragraph to improve clarity, as follows: “The included studies comprised both preclinical and clinical investigations exploring epigenetic alterations in GBM as therapeutic targets. Charted data (Tables 1-3) reflects a diverse range of epigenetic mechanisms, including DNA methylation, histone modifications, and ncRNAs. Therapeutic approaches varied, encompassing small-molecule inhibitors, RNA-based therapies, and gene editing strategies. Most included studies were conducted in preclinical settings using established GBM cell lines, patient-derived xenografts or animal models, with a smaller proportion representing early-phase or small-scale clinical trials.” (page 5, lines 154-161).

Comment 6: Some sections (e.g., histone modification, EZH2, HDACs) are extremely dense and could benefit from condensation. Consider focusing more sharply on mechanistically distinct, translationally relevant findings and minimizing repetition across subsections. Several signaling axes are re-explained in multiple parts of the manuscript.

Response 6: We concur with the reviewer’s observation and have revised the manuscript accordingly. The main alterations and deletions made include: “part of the transmembrane RTK family” (page 8, line 306); “NF-κB activation is strongly linked to aggressive GSC phenotypes, driving processes such as EMT, invasion, angiogenesis, and TMZ resistance.” (page 18, lines 788-789); “a critical factor for maintaining GSC stemness, highlighting its importance as a therapeutic target” (page 20, lines 871-872); “Vorinostat is an HDACi, FDA-approved for cutaneous T cell lymphoma, that has shown modest effectiveness as a standalone therapy in recurrent GBM [125].” (page 20, lines 912-913).

Comment 7: Given the complexity of the content, the manuscript would greatly benefit from: A summary table of the major epigenetic targets, associated pathways, and therapeutic agents discussed and/or a graphical abstract or figure showing the integration of epigenetic modifications with therapeutic strategies in GBM.

Response 7: We thank the reviewer for the observation and suggestion. As mentioned in Response 4, the manuscript already includes one table summarizing the epigenetic mechanisms and associated pathways (Table 1, pages 31-33), and another table summarizing the therapeutic agents and their targets (Table 2, pages 33-34). Additionally, a graphical abstract has been provided (Figure 1, page 2), illustrating the eight domains discussed in the manuscript along with key examples for each epigenetic subtopic.

Comment 8: Revise some verbose or speculative statements, e.g., "holds the potential to redefine GBM treatment paradigms..." should be made more evidence-based.

Response 8: As advised, we have made the necessary changes to improve clarity on this issue by rewriting the following sentences: “Leveraging insights from epigenetics, coupled with precision medicine tools like omics-driven biomarker identification, may enable the development of more effective and personalized treatment strategies for GBM.” (page 3, lines 100-103); “Therefore, MCM4 has been identified as a potential therapeutic target and prognostic marker in GBM, with reported associations to immune regulation, tumor progression, and drug sensitivity [120].” (page 19, lines 859-861); “Advances in drug delivery, target specificity, and overcoming resistance mechanisms, combined with interdisciplinary approaches, may enhance the effectiveness of epigenetic therapies in improving outcomes for GBM patients.” (page 27, lines1233-1236).

Comment 9: Ensure consistency in terminology (e.g., GBM vs. glioblastoma).

Response 9: We are grateful for this suggestion, which has helped improve the clarity of our work. The term “glioblastoma” has been replaced with the abbreviation “GBM” (although not highlighted in yellow”) and “glioblastoma” has been added to the abbreviations table, next to “glioblastoma multiforme” (page 28).

Comment 10: To strengthen the review’s completeness, please consider citing the following article when discussing molecular predictors and in glioblastoma:

   Saaid et al.: https://doi.org/10.3390/genes13081439. This paper contextualizes epigenetic and genetic predictors in GBM, including MGMT and IDH-related methylation profiles, and should be cited when discussing the predictive relevance of DNA methylation and IDH mutations (e.g., lines 88–95 or 264–266).

   Along with citation 6 at line 99 I would cite this review (https://doi.org/10.3390/biomedicines11061520) regarding immunotherapy and oncolytic viruses in glioma.

Response 10: We acknowledge the reviewer’s suggestion and appreciate the opportunity to refine our work. The first article (https://doi.org/10.3390/genes13081439) has been added to the Introduction (page 3, line 93), as suggested. Although the second article (https://doi.org/10.3390/biomedicines11061520) presents relevant ideas that align with the content discussed in section 4.7, “Immunomodulation via Epigenetics,” it is a review article and therefore does not meet the inclusion criteria for our study.

Comment 11: Define all abbreviations at first mention in each section (e.g., PRMT, GSC, SE).

Response 11: We appreciate the reviewer’s comment. As stated in the IJMS instructions for authors, and in agreement with the suggestion made, we have revised the manuscript to ensure that all abbreviations are defined at first mention in the text and listed in the abbreviations table.

Comment 12: Add page/figure/table numbers to improve navigation through the manuscript.

Response 12: We thank the reviewer for this comment. We would like to note that this information was already included in the original version – Image 1 (page 2), Image 2 (page 3), Image 3 (page 5), and Tables 1–3 (pages 31–36). The page numbers appear in the upper right corners of each page, as part of the IJMS Word template used.

4. Response to Comments on the Quality of English Language

Point 1: The English is fine and does not require any improvement.

Response: Thank you for your positive evaluation.

5. Additional clarifications

We hope that the manuscript may now be suitable for publication. We confirm that neither the manuscript nor any parts of its content are currently under consideration or published in another journal. All authors have approved the revised version of the manuscript and agree with its submission to IJMS. Thank you for kindly considering our work.

Reviewer 2 Report

Comments and Suggestions for Authors

See atached comments

Author Response

3. Point-by-point response to Comments and Suggestions for Authors

Comment 1: Meleiro and Henrique surveyed the literature concerning epigenomic alterations as well as related therapeutic approaches in glioblastoma. This is a timely choice of an important topic. The Introduction is appropriate. The Methods section clearly describes the applied standard scoping review methodology, and Figure 3 well reflects the comprehensive strategy of literature survey and study inclusion. The Results section briefly details how the authors arrived to the inclusion of 273 studies, also depicted in the PRISMA-ScR flow diagram (Figure 3), and lists “Epigenetic related pathophysiology and potential therapeutic targets in GBM (specified in Figure 1, and Suppl Table 1), “Main search results: epigenetic-related molecules and drugs in GBM” (specified in Suppl Table 2), and “Most relevant clinical trials incorporating epigenetics in GBM” (specified in Suppl Table 3). Finally, an explanation is provided regarding the synthesis of findings focusing on translational relevance, concept of prioritization and the main focus on papers within the period of 2021 – 2024, also included in Ref #15. The main part of the survey can be found in the Discussion section with eight subtitles. The list of references includes 274 papers, a large

segment of the recent and relevant literature. The manuscript reflects a thorough, systematic presentation of the selected literature, giving a comprehensive molecular overview of current knowledge regarding GBM pathogenesis and experimental treatment strategies.

The strength of the paper: The topic choice is excellent. The scientific aspects of the work are both very comprehensive and elaborate. The methodology and the results are clearly described and discussed. English language is very well used. The Figures and the Supplementary tables are very informative.

Response 1: We greatly appreciate your attention to detail and the helpful remarks provided throughout the review.

Comment 2: A potential weakness of the paper: A large amount of molecular information is listed in a way that may not be easily absorbed and synthetized by readers working outside of the field. The cause of this relative weakness is related to a lack of driving concept for the data presentation. This reviewer feels that the inclusion of a figure with a simplified depiction of main molecular pathways (and their key elements) with identified epigenetic modifications during GBM pathogenesis, which may also serve as potential treatment targets, would make comprehension of the information easier.

Response 2: Thank you for your thoughtful feedback and for your engagement to our work. In response to your comment, we have designed a new image – “Figure 4. Simplified Overview of Epigenetic and Signaling Pathways in GBM Pathogenesis and Therapeutic Targeting” (page 7, lines 246–249). We hope this new figure meets your expectations and enhances the clarity and understanding of our work.

Comment 3: A suggestion: The title of Figure 1 may be modified from “An overview of

altered epigenetic mechanisms in glioblastoma” to “An overview of altered epigenetic mechanisms and emerging therapeutic interventions in glioblastoma.”

Response 3: Thank you for this insightful observation. We fully agree and have made the suggested change (page 2, lines 41-42).

Comment 4: An observation from the reviewer: It appears that the applied deduplication

method excluded some original papers that only partially overlapped with the one ultimately included, and therefore, some potentially notable research data were omitted from the text (and also from Suppl Table 1).

Response 4: We acknowledge the importance of this point and recognize that it may not have been sufficiently explained in the manuscript. To enhance clarity, we have restructured the relevant paragraph as follows: “Duplicate records were removed using EndNote, initially by matching records with the same title and author published in the same year, followed by a second pass using title, author, and journal. This process identified 93 duplicates. An additional 78 duplicates were identified through manual review. A total of 171 duplicates were removed before screening.” (page 4, lines 128–132). We believe that this method did not exclude partially overlapping articles.

Comment 5: Overall, this is a highly informative and invaluable work that could be important not only for basic science researchers but also for clinicians in the field.

Response 5: Thank you for your careful review and for contributing to the improvement of this study.

4. Response to Comments on the Quality of English Language

Point 1: The English is fine and does not require any improvement.

Response: Thank you for your positive evaluation.

5. Additional clarifications

We hope that the manuscript may now be suitable for publication. We confirm that neither the manuscript nor any parts of its content are currently under consideration or published in another journal. All authors have approved the revised version of the manuscript and agree with its submission to IJMS. Thank you for kindly considering our work.

Round 2

Reviewer 1 Report

Comments and Suggestions for Authors

accept

Author Response

1. Summary

Thank you very much for taking the time to review this manuscript. Please find our detailed responses below and note that the corresponding corrections have been highlighted in yellow in the re-submitted files.

2. Point-by-point response to Comments and Suggestions for Authors

Comment 1: While the review is thorough, covering literature extensively up to late 2024, I believe its timeliness and scope could be further enhanced by incorporating some of the very latest references and a brief discussion on a couple of rapidly evolving areas.

Response 1: We are grateful for this suggestion, which has helped us improve the timeliness and relevance of our work. All suggested articles were retrieved, analyzed, and assessed for eligibility according to the inclusion and exclusion criteria previously defined in the Methods section: “All relevant preclinical studies and clinical trials – single-arm or double-arm, including both randomized and non-randomized controlled trials – addressing epigenetic alterations in GBM as therapeutic targets were eligible for inclusion. Studies were excluded if they did not clearly define their methods and/or results. Only studies published in English and within ten years (2014-2024) were considered. Editorials, conference abstracts, books or book chapters, case reports, case series, literature reviews, meta-analyses, and preprints were excluded.” (page 4, lines 112-118).

Comment 2: Specifically, it might be beneficial to consider recent advances in the general role of epigenetic modifications, perhaps within the Introduction or Discussion sections, of the most recent (e.g., early 2025) high-impact findings regarding the fundamental roles of epigenetic modifications both broadly in cancer (doi: 10.1111/j.1399-0004.2011.01809.x; https://doi.org/10.1186/s12943-020-01197-3; doi: 10.1136/jclinpath-2020-206633; doi: 10.1016/j.tranon.2023.101821) and with any new specific insights for GBM (https://doi.org/10.3390/ijms26073368; doi: 10.1111/bpa.13334; doi: 10.20517/cdr.2024.157; doi: 10.3390/cells14070494; https://doi.org/10.3389/freae.2025.1519449).

Response 2: We appreciate the insightful comment provided. After careful consideration, the following paragraph was added to Section 4.7, based on the article identified by DOI:  10.1111/bpa.13334: “Peripheral immune cell counts have been linked to tumor cell states and immune infiltration, being considered as biomarkers in GBM when integrated with DNA methylation-based subclassification. Elevated neutrophil counts are associated with worse overall survival in newly diagnosed GBM, while decreases in lymphocytes, monocytes, and platelets correlate with progression and poor outcomes in recurrent disease. These findings support a subclass-specific approach to immunotherapy in GBM and underscore the potential of peripheral blood profiling, when interpreted through an epigenetic lens, to improve prognostication and patient selection [142].” (page 23, lines 1033-1040).

The paper identified by DOI: 10.1136/jclinpath-2020-206633 focuses on bladder cancer rather than GBM and therefore does not meet the inclusion criteria. The remaining articles (DOI:  10.1111/j.1399-0004.2011.01809.x; https://doi.org/10.1186/s12943-020-01197-3; DOI: 10.1016/j.tranon.2023.101821; https://doi.org/10.3390/ijms26073368; DOI: 10.20517/cdr.2024.157; DOI: 10.3390/cells14070494; https://doi.org/10.3389/freae.2025.1519449) are literature reviews and therefore fall under the exclusion criteria.

Comment 3: Also, slightly expanding the discussion on how epigenetic alterations might regulate autophagic pathways in GBM, or conversely, how autophagy might influence the epigenetic landscape, and the therapeutic implications of this crosstalk, could be a valuable addition. Recent literature is increasingly exploring this nexus, and including the latest perspectives could highlight novel therapeutic vulnerabilities (https://doi.org/10.3892/ol.2017.7446; doi: 10.1186/s12967-024-06063-0; https://doi.org/10.3390/cells13161332; doi: 10.3390/cancers15092622). Incorporating these very recent perspectives would further solidify the review's position as a current and exhaustive resource.

Response 3: Thank you for pointing this out. To further explore the interplay between autophagy and epigenetics in GBM, the following text was added, based on the article identified by https://doi.org/10.3892/ol.2017.7446: “Bevacizumab, a monoclonal antibody targeting vascular endothelial growth factor (VEGF)-A, inhibits the AKT/mTOR pathway, inadvertently activating autophagy as a survival mechanism. While bevacizumab effectively reduces proliferation and enhances apoptosis by modulating pro- and anti-apoptotic protein levels, its therapeutic impact is limited by this compensatory autophagic response. Blocking autophagy with chloroquine significantly amplifies bevacizumab-induced apoptosis, suggesting that autophagy enables GBM cells to tolerate anti-angiogenic stress. These findings highlight the interplay between the AKT/mTOR pathway and autophagy in driving resistance and point to autophagy inhibition as a promising strategy to enhance bevacizumab efficacy in GBM treatment [77].” (page 12, lines 496-505).

The remaining articles (doi: 10.1186/s12967-024-06063-0; https://doi.org/10.3390/cells13161332; doi: 10.3390/cancers15092622) are literature reviews and therefore fall under the exclusion criteria.

Comment 4: Nonetheless, the current manuscript stands as a significant and commendable contribution to the field.

Response 4: Thank you for your positive evaluation.

3. Additional clarifications

We hope that the manuscript may now be suitable for publication. We confirm that neither the manuscript nor any parts of its content are currently under consideration or published in another journal. All authors have approved the revised version of the manuscript and agree with its submission to IJMS. Thank you for kindly considering our work.
